# Swarm Reinforcement Learning for Adaptive Mesh Refinement

**Niklas Freymuth**[1]* **Philipp Dahlinger**[1] **Tobias Würth**[2] **Simon Reisch**[1]
**Luise Kärger**[2] **Gerhard Neumann**[1]
[1]Autonomous Learning Robots, Karlsruhe Institute of Technology, Karlsruhe
[2]Institute of Vehicle Systems Technology, Karlsruhe Institute of Technology, Karlsruhe

## Abstract

Adaptive Mesh Refinement (AMR) enhances the Finite Element Method, an important technique for simulating complex problems in engineering, by dynamically refining mesh regions, enabling a favorable trade-off between computational speed and simulation accuracy. Classical methods for AMR depend on heuristics or expensive error estimators, hindering their use for complex simulations. Recent learning-based AMR methods tackle these issues, but so far scale only to simple toy examples. We formulate AMR as a novel Adaptive Swarm Markov Decision Process in which a mesh is modeled as a system of simple collaborating agents that may split into multiple new agents. This framework allows for a spatial reward formulation that simplifies the credit assignment problem, which we combine with Message Passing Networks to propagate information between neighboring mesh elements. We experimentally validate our approach, Adaptive Swarm Mesh Refinement (ASMR), on challenging refinement tasks. Our approach learns reliable and efficient refinement strategies that can robustly generalize to different domains during inference. Additionally, it achieves a speedup of up to 2 orders of magnitude compared to uniform refinements in more demanding simulations. We outperform learned baselines and heuristics, achieving a refinement quality that is on par with costly error-based oracle AMR strategies.

## 1 Introduction

The Finite Element Method (FEM) is a widely used numerical technique in engineering and applied sciences for solving complex partial differential equations [1, 2, 3, 4]. The method discretizes the continuous problem domain into smaller, finite elements, allowing for an efficient numerical solution. A key aspect of the FEM for complex systems is Adaptive Mesh Refinement (AMR), which dynamically refines regions of high solution variability, allowing for a favorable trade-off between computational speed and simulation accuracy [5, 6, 7]. As problems in engineering grow more complex, the FEM and especially Adaptive Mesh Refinement (AMR) techniques have become increasingly important tools in providing computationally tractable yet precise solutions. Applications of AMR include fluid dynamics [8, 9, 10, 11, 12, 13], structural mechanics [14, 15, 16, 17], and astrophysics [18, 19, 20]. Yet, classical approaches for AMR usually rely on either problem-dependent or more general but potentially suboptimal error indicators, or require expensive error estimates [21, 22, 23, 24, 25, 26, 12]. In either case, they can be cumbersome to use in practice.

To address this issue, we formalize AMR as a Reinforcement Learning (RL) [27] problem. Following previous work [28, 29, 30], each refinement step encodes the state of the current simulation as local observations that we feed to RL agents, who then determine which elements of a mesh to refine. However, previous work has issues with scalability due to an expensive inference process [28],

---

*correspondence to `niklas.freymuth@kit.edu`

37th Conference on Neural Information Processing Systems (NeurIPS 2023).

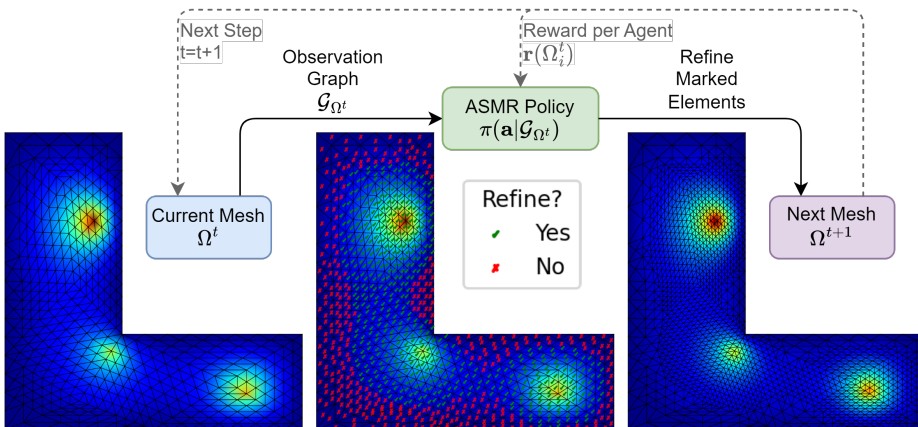

Figure 1: Given a mesh $\Omega^t$, an observation graph $\mathcal{G}_{\Omega^t}$ encodes the elements as graph nodes and the neighborhood between elements as edges. The graph is given to a learned policy $\pi$, which marks mesh elements for refinement. A remesher refines the mesh, and spatial rewards $\mathbf{r}(\Omega_i^t)$ are calculated for all agents $i$ based on the quality of the refinement. This process is iterated for several steps until the mesh is fully refined.

misaligned objectives and high variance in the state transitions [30], and noisy reward signals [29]. To mitigate these shortcomings and scale to more complex problems, we formulate AMR as a Swarm RL [31, 32] problem. We extend the Swarm RL framework to per-agent rewards, a shared observation space, and the option of splitting agents into new agents as required in the AMR process. Additionally, we introduce a novel spatial reward formulation that provides a dense reward signal for the refinement of each mesh element, simplifying the credit assignment problem for swarm systems. Our policy is based on Message Passing Networks (MPNs) [33], a class of Graph Neural Networks (GNNs) [34, 35, 36, 37] that has proven to be effective for physical simulations [38, 39]. The resulting method, Adaptive Swarm Mesh Refinement (ASMR), consistently produces highly refined meshes with thousands of elements while applying to arbitrary Partial Differential Equations (PDEs). A high-level overview is given in Figure 1.

Experimentally, we show the effectiveness of our approach on a suite of PDEs that require complex and challenging refinement strategies, including a non-stationary heat diffusion problem and a linear elasticity task. We implement our tasks as OpenAI gym [40] environments. The experiments use static meshes with conforming triangular elements and corresponding h-adaptive refinements [41, 42] due to their importance in engineering applications [43, 44, 45, 46]. Here, accurate meshes require multiple precise refinement steps and thousands of elements. We implement and compare to current state-of-the-art RL methods for AMR [28, 29, 30] that have been shown to work well on dynamic tasks where shallow mesh refinement and coarsening is sufficient[1]. We observe that these methods struggle with finer static meshes, whereas ASMR yields stable and consistent refinements across tasks. To further evaluate our method's effectiveness, we compare it to the popular Zienkiewicz-Zhu Error Estimator (*ZZ Error*) estimate and a traditional AMR heuristic requiring oracle error estimates. ASMR not only outperforms both learned and traditional methods that lack oracle data but also achieves performance comparable to oracle-based heuristics. Furthermore, ASMR is 2 to 100 times faster than computing the uniform mesh on which the oracle information is based, and demonstrates robust generalization capabilities across different domains and initial conditions. We conduct a series of ablations to show which parts of the approach make it uniquely effective, finding that spatial rewards per agent are preferable to a shared global reward signal for all agents.

To summarize our contributions, we (1) propose a novel Markov Decision Process (MDP) formulation that naturally integrates local rewards for swarms where agents may split into new agents over time; (2) combine this formulation with MPNs and a novel spatial reward formulation to reliable and efficiently scale learned AMR to static meshes with thousands of elements on multiple levels of refinement; (3) showcase our approach's effectiveness on a suite of PDEs with challenging refinement

---

[1]We publish the first codebase on RL for AMR, including all methods and tasks presented in this paper, to facilitate research in this direction. The code is available at `https://github.com/NiklasFreymuth/ASMR`.

problems. Our method surpasses state-of-the-art RL methods and the popular *ZZ Error* heuristic, and achieves refinement quality comparable to oracle-based AMR strategies, all without requiring costly error estimates during inference.

## 2   Related Work

**Learned Physics Simulation.** A considerable body of work deals with directly learning to simulate physical systems with neural networks. These approaches typically learn from data generated by some underlying ground-truth simulator and train the network to predict the (change in) quantities of interest during a simulation. Such learned physics simulators are fully differentiable and often orders of magnitude faster than their classical counterparts [35, 38], lending them to use cases such as Inverse Design [47, 48, 49]. Researchers have developed simulators based on simple feed-forward networks [50, 51] and Convolutional Neural Networks [52, 53, 54, 55, 56, 57, 58, 59, 60]. Closely related to our method are Graph Network Simulators (GNSs) [33, 38, 61, 62, 63, 64, 39, 65], which utilize GNNs to encode physical problems as a graph on which to compute quantities of interest per node. Here, a recent method [66] jointly learns a GNS and an AMR strategy on mesh edges to allow for simulation on different resolutions.

Physics-Informed Neural Networks [67, 68, 69] are mesh-free methods designed to directly train neural networks to satisfy the governing equations of a physical system. They share the goal of using deep neural networks to solve PDEs, yet differ in their approach in that they directly approximate the equations rather than providing a mesh for a classical solver. Thus, AMR strategies provide a more robust, flexible, and risk-averse approach to solving complex physics problems that demand high precision and accuracy [11, 70, 13]. As Physics-Informed Neural Networks also operate on geometric domains, they have been extended to GNN architectures [71, 72, 73]. In this work, we do not learn to solve a system of equations directly, but rather propose an efficient mesh refinement for a classical solver.

**Supervised Learning for AMR.** Applications of supervised learning for AMR include directly calculating an error per mesh element with a Multilayer Perceptron (MLP) [11] and predicting mesh densities from domain images [70]. Additionally, recurrent networks have been used to find optimal marking strategies for second-order elliptical PDEs [74]. Another body of work speeds up the computation of Dual Weighted Residual [75, 76] error estimators by substituting expensive parts of the procedure with neural networks. Here, recent methods [77, 78] consider learning a metric tensor from solution information that can then be used in existing refinement procedures [24]. Other approaches [79, 13] employ neural networks to solve the strong form of the adjoint problem and use hand-crafted features to compute error estimates directly. We instead leverage the fact that RL can optimize non-differentiable rewards, enabling us to directly learn a refinement strategy in an iterative manner instead of learning error estimators or other specific facets of AMR.

**Reinforcement Learning for AMR.** Current research in Reinforcement Learning for AMR involves various approaches, such as optimizing the positions of mesh elements [80], predicting a global threshold for heuristic-based refinement using existing error estimates [81], and generating quadrilateral meshes by iteratively extracting elements from the problem domain [82].

We instead directly manipulate the mesh elements themselves. This approach presents a unique challenge in that the size of the observation and action spaces is constantly changing during refinement. Existing methods [28, 29, 30] derive their observation spaces from the mesh geometry and the solution computed on the mesh. These methods are generally designed for non-stationary PDEs and thus include mesh coarsening operations that are required for time-dependent problems. While our method can easily be extended to coarsening and time-dependent problems, this work instead considers static meshes and mostly stationary problems due to their prevalence in engineering [43, 44] . Here, the challenge lies in finding multiple levels of accurate refinements rather than shallow or local time-dependent refinement and coarsening. The earliest of these methods [28] treats the entire mesh as an action and observation space for a *Single Agent*, which uses a GNN-based policy to provide a categorical action that selects a single element for refinement. *Single Agent* demands solving the system of equations after each refinement step, significantly increasing inference time while reducing the amount of information per environment sample. Another approach [30] iteratively selects a random element during training and uses an MLP policy to determine its marking based on local and global features. During inference, the method performs a *Sweep* for all mesh elements in parallel.

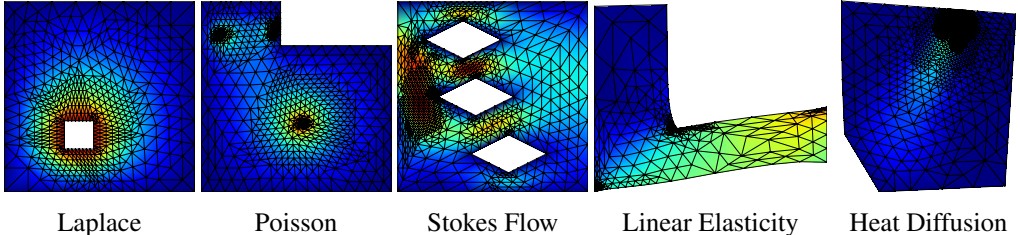

| Laplace | Poisson | Stokes Flow | Linear Elasticity | Heat Diffusion |

Figure 2: Exemplary ASMR refinements. The heatmaps represent the normalized quantities of interest. ASMR provides complex and accurate refinements for different tasks. From left to right: Laplace's equation requires a homogeneous refinement near the source at the inner boundary. For Poisson's equation, a multi-modal load function causes multiple distinct regions of interest. The Stokes flow uses more complex shape functions and requires high precision near the inlet on the left and the rhomboid holes. For the linear elasticity task, we consider both the deformation and the resulting stress as quantities of interest. The heat diffusion task requires accurate refinements on the path of the moving heat source to predict an accurate solution of the final step.

This procedure speeds up inference but causes a misalignment in the environment transition between training and inference [29]. Additionally, during training, the agent is randomly assigned a new element after each action, leading to high variance in the state transitions.

Most similar to our method are Value Decomposition Graph Networks (*VDGN*) [29] which frame AMR as a cooperative multi-agent problem by setting a maximum refinement depth and number of agents. *VDGN* employs a Value Decomposition Network [83] to circumvent the posthumous credit assignment problem [84] of vanishing agents. Though theoretically efficient in training and inference, the method's performance depends on the quality of the value decomposition, which becomes more difficult for larger meshes. In summary, existing RL methods do not utilize the spatial nature of AMR and thus only scale to either simple or comparatively shallow refinements. We instead formulate AMR as a Swarm Reinforcement Learning problem with spatial rewards, naturally integrating changing observation and action spaces while also providing a strong feedback signal to all agents.

## 3 Adaptive Swarm Mesh Refinement

In the following, we introduce the individual components of ASMR, including our novel Adaptive Swarm Markov Decision Process (ASMDP) and spatial reward function. We consider each element $\Omega_i^t \in \Omega^t$ of a mesh $\Omega^t$ to be an agent in a swarm system. The agent's state is its position in the mesh, as well as boundary conditions and other PDE-dependent quantities. The agent's observation consists of a local view of a graph $\mathcal{G}_{\Omega^t}$ where each node represents a mesh element and each edge the neighborhood of two elements. We train a simple MPN-based policy $\pi(\mathbf{a}|\mathcal{G}_{\Omega^t})$ that computes a joint action vector $\mathbf{a} \in \mathcal{A}^N$ for each mesh element by passing messages along the observation graph, as detailed in Appendix A. The action vector is used for refinement, and the process is repeated with the refined mesh for a given number of steps. Since our policy uses a GNN, it is equivariant to permutation and can handle varying numbers of agents by construction. Figure 1 provides a schematic overview of our method.

**Adaptive Swarm Markov Decision Process.** We adapt the SwarMDP framework [31, 32] to incorporate action and observation spaces of changing size as necessary for AMR, agent-wise rewards and mappings between agents over time. The resulting framework is conceptually simpler than, e.g., a decentralized partially obeservable MDP with dummy states [29], makes the permutation-equivariance of the agents explicit and naturally integrates both the agent-dependent reward and the mapping between agents. Formally, we define an Adaptive Swarm Markov Decision Process (ASMDP) as a tuple $\langle \mathbb{S}, \mathbb{O}, \mathbb{A}, T, \mathbf{r}, \xi, \mathbf{M} \rangle$. Here, $\mathbb{S}$ is the state space of the system, $\mathbb{O}$ is the space of observations for this state space, and $\mathbb{A}$ is the action space for the system of agents. Let $\mathcal{S}^N \subset \mathbb{S}$, $\mathcal{O}^N \subset \mathbb{O}$, $\mathcal{A}^N \subset \mathbb{A}$ denote the subsets of the state, observation, and action spaces with exactly $N$ agents. The transition function $T : \mathcal{S}^N \times \mathcal{A}^N \rightarrow \mathcal{S}^K$ maps to a new system state with a potentially different number of agents, and $\mathbf{r} : \mathcal{S}^N \times \mathcal{A}^N \rightarrow \mathbb{R}^N$ is a per-agent reward function. The observation graph of the agents is calculated from their states via the observation function $\xi : \mathcal{S}^N \rightarrow \mathcal{O}^N$. To

accommodate changing numbers of agents throughout an episode, we define an agent mapping $\mathbf{M}^t \in [0,1]^{N \times K}$ with $\sum_i \mathbf{M}^t_{ij} = 1$ for all $j = 1, \ldots, K$ that specifies how agents evolve at time step $t$. Each entry $\mathbf{M}^t_{ij}$ describes whether agent $i$ at step $t$ progresses into agent $j$ at step $t + 1$. The influence of each agent at step $t$ on the reward, in terms of all successor agents it is responsible for up to step $t + k$, can then be computed via the matrix multiplication $\mathbf{M}^{t,k} := \mathbf{M}^t \mathbf{M}^{t+1} \ldots \mathbf{M}^{t+k-1}$.

The usual objective in RL is to find a policy $\pi : \mathcal{O}^N \times \mathcal{A}^N \to [0,1]$ that maximizes the return, i.e., the expected discounted cumulative future reward $J^t := \mathbb{E}_{\pi(\mathbf{a}|\xi(\mathbf{s}))}\left[\sum_{k=0}^{\infty} \gamma^k r(\mathbf{s}^{t+k}, \mathbf{a}^{t+k})\right]$ for a discount factor $\gamma$ and scalar reward $r(\mathbf{s}^{t+k}, \mathbf{a}^{t+k})$ at step $t + k$. Adapting this to varying numbers of agents within a single episode, as necessary for e.g., the refinement of mesh elements, yields

$$J^t_i := \mathbb{E}_{\pi(\mathbf{a}|\xi(\mathbf{s}))}\left[\sum_{k=0}^{\infty} \gamma^k (\mathbf{M}^{t,k} \mathbf{r}(\mathbf{s}^{t+k}, \mathbf{a}^{t+k}))_i\right], \tag{1}$$

for agent $i$ at step $t$. Intuitively, this return represents the discounted sum of rewards of all agents that agent $i$ is responsible for. We set $V_i(\mathbf{s}^t) = \mathbf{r}(\mathbf{s}^t, \mathbf{a}^t)_i + \gamma \sum_j \mathbf{M}^t_{ij} V_j(T(\mathbf{s}^t, \mathbf{a}^t))$ with $\mathbf{a} \sim \pi(\xi(\mathbf{s}))$ for training the value function and derive the targets for $Q$-functions analogously.

**Agents and Observations.** Given a domain $\Omega$ and a mesh $\Omega^t := \{\Omega^t_i \subseteq \Omega | \dot{\bigcup}_i \Omega^t_i = \Omega\}$, we view each mesh element $\Omega^t_i$ as an agent. Each element's action space comprises a binary decision to mark it for refinement. These markings are provided to a remesher, which refines all marked elements, yielding a finer mesh $\Omega^{t+1} = \{\Omega^{t+1}_j\}_j$. Here, $\Omega^{t+1}_j := \Omega^t_i$ for no refinement and $\Omega^{t+1}_j \subsetneq \Omega^t_i$ with $\dot{\bigcup}_j \Omega^{t+1}_j = \Omega^t_i$ if $\Omega^t_i$ is refined. The remesher may also refine unmarked elements to assert a conforming solution [41], i.e., to make sure that elements of the mesh align with each other at the boundaries and interfaces to ensure continuity of solution variables between adjacent elements. We define the mapping for an agent to its successor agents as the indicator function $\mathbf{M}^t_{ij} := \mathbb{I}(\Omega^{t+1}_j \subseteq \Omega^t_i)$. In other words, an agent maps to all future agents that it spawns, or equivalently, an element is responsible for all sub-elements that it refines into over time. While we focus on mesh refinement in this work, this mapping can be extended to coarsening by setting, e.g., $\mathbf{M}^t_{ij} := \mathbb{I}(\Omega^{t+1}_j \subseteq \Omega^t_i) + \left(\mathbb{I}(\Omega^t_i \subsetneq \Omega^{t+1}_j)/\left(\sum_k \mathbb{I}(\Omega^t_k \subsetneq \Omega^{t+1}_j)\right)\right)$.

For encoding the observations, we use an observation graph $\mathcal{G}_{\Omega^t} = \mathcal{G} = (\mathcal{V}, \mathcal{E}, \mathbf{X}_\mathcal{V}, \mathbf{X}_\mathcal{E})$, which is a bidirectional directed graph with mesh elements as nodes $\mathcal{V}$ and their neighborhood relation as edges $\mathcal{E} \subseteq \mathcal{V} \times \mathcal{V}$. Node and edge features of dimensions $d_\mathcal{V}$ and $d_\mathcal{E}$ are given as $\mathbf{X}_\mathcal{V} : \mathcal{V} \to \mathbb{R}^{d_\mathcal{V}}$ and $\mathbf{X}_\mathcal{E} : \mathcal{E} \to \mathbb{R}^{d_\mathcal{E}}$. Further details can be found in Appendix B.

**Reward.** A good refinement strategy trades off the accuracy of the solution of the mesh $\Omega^t$ with its total number of elements $\Omega^t_i \in \Omega^t$. We define an error per element as the difference in the solution of this element compared to a solution using a fine-grained reference mesh $\Omega^*$ [28]. We consider $\Omega^*$ to be optimal, but very slow to compute due to a large number of elements. However, we only require the reference mesh $\Omega^*$ for the reward calculation, not during inference. For each element $\Omega^t_i$ we then integrate over the evaluated differences of all midpoints $p_{\Omega^*_m} \in \Omega^*_m$ of reference elements $\Omega^*_m$ that fall into it, scaling each by the area $\text{Area}(\Omega^*_m)$ of its respective element. This procedure results in an error estimate

$$\hat{\text{err}}(\Omega^t_i) \approx \sum_{\Omega^*_m \subseteq \Omega^t_i} \text{Area}(\Omega^*_m) \left|u_{\Omega^*}(p_{\Omega^*_m}) - u_{\Omega^t}(p_{\Omega^*_m})\right|, \tag{2}$$

where $u_{\Omega^*}$ denotes the solution on the fine mesh and $u_{\Omega^t}$ the solution on the current mesh. We note that this error estimate can be efficiently calculated using a $k$-d tree [85] and that it is generally applicable for a large range of PDEs. Problem-specific error estimates may be used instead to include domain knowledge. To get an error estimate that is consistent across different geometries, we normalize the error with the total error of the elements of the initial mesh $\Omega^0$, i.e., $\text{err}(\Omega^t_i) = \hat{\text{err}}(\Omega^t_i)/\sum_{\Omega^0_j \in \Omega^0} \hat{\text{err}}(\Omega^0_j)$. We then formulate a local reward per element as

$$\mathbf{r}(\Omega^t_i) := \frac{1}{\text{Area}(\Omega^t_i)}\left(\text{err}(\Omega^t_i) - \sum_j \mathbf{M}^t_{ij}\text{err}(\Omega^{t+1}_j)\right) - \alpha\left(\sum_j \mathbf{M}^t_{ij} - 1\right), \tag{3}$$

where $\alpha$ is a hyperparameter that penalizes adding new elements.

This reward function evaluates whether a refinement decreases the overall error by enough to justify the extra resources required, with a reward of 0 for unrefined elements. Thus, the reward maximizes

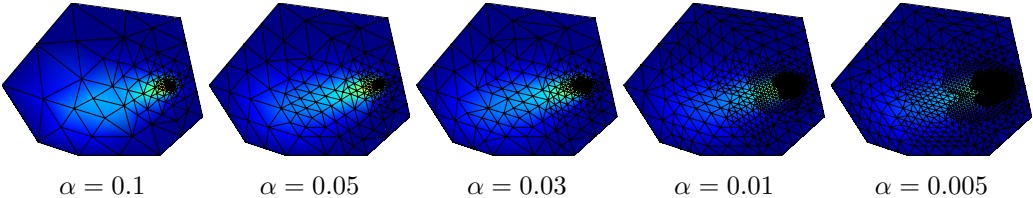

$$\alpha = 0.1 \qquad \alpha = 0.05 \qquad \alpha = 0.03 \qquad \alpha = 0.01 \qquad \alpha = 0.005$$

Figure 3: Final refinements of ASMR on a Heat Diffusion problem for different values of the element penalty $\alpha$. All refinements focus on the relevant parts of the problem, and lower element penalties lead to more fine-grained meshes.

error reduction, rather than simply encouraging a refinement of areas with a high existing error regardless of the resulting mesh improvement. Further, incorporating a novel area scaling term encourages the policy to focus on smaller elements with a high potential reduction in error rather than larger elements with a low average error reduction, up to some threshold depending on the element penalty $\alpha$. We find that the combination of a local formulation and the area scaling term allows for a simpler credit assignment for the RL agents, as it ensures that each agent gets rewarded for its own actions and that rewards of elements of different sizes are on the same scale. Similarly, the area scaling term of the reward effectively cancels out the area of the integration points in Equation 2, causing the policies optimized on this reward to implicitly minimize the maximum remaining error of the mesh, while also making sure that the mean error stays sufficiently low. We compare this to directly minimizing the maximum error in Appendix C.2.

Since the effects of mesh refinement can be non-local for elliptical PDEs, we optimize the average of the local and global returns, i.e.,

$$J_i^{t'} = \frac{1}{2}J_i^t + \frac{1}{2}J^t, \tag{4}$$

where $J_i^t$ is the return of agent $i$ at step $t$ as shown in Equation 1, and $J^t$ is the global return calculated using the average reward $r = \frac{1}{N}\sum_j \mathbf{r}_j$. In multi-quantity systems of equations, it is important for the mesh to be suitable for all the quantities of interest. For this, we calculate individual errors $\mathrm{err}^d(\Omega_i^t)$ for each solution dimension and then use a norm or a convex sum of these as the overall error, depending on the application.

## 4 Experiments

**Setup.** All learned methods are trained on 100 PDEs and their corresponding initial and reference meshes $\Omega^0$, $\Omega^*$ to limit the number of required reference meshes during training. We experiment with 10 different target mesh resolutions per method to produce a wide range of solutions, as detailed in Appendix F.3. We repeat each experiment for 10 random seeds and always report the average performance on 100 randomly sampled but fixed evaluation PDEs. These PDEs are disjoint from the training PDEs, and both sets of PDEs consist of randomly sampled domains as well as boundary and initial conditions as detailed below. Details on the setup and the computational budget for our experiments are provided in Appendix C.1. The reference mesh $\Omega^*$ is created by uniformly refining the initial mesh 6 times. An environment episode consists of drawing one of the 100 training PDE without replacement, and iteratively refining the coarse initial mesh $\Omega^0$ a total of $T = 6$ times unless mentioned otherwise. Since the maximum number of elements scales exponentially with the refinement depth, we additionally evaluate a simpler task setup with $T = 4$ refinements to roughly replicate the task complexity of existing work [28, 63, 29]. We experiment with Deep Q-Network (DQN) [86, 87] as an off-policy and Proximal Policy Optimization (PPO) [88] with discrete actions as an on-policy RL algorithm for all RL-based methods.

We evaluate mesh quality by calculating the squared error at each point in the high-resolution reference $\Omega^*$, i.e., as $\sum_{\Omega_m^* \in \Omega^*} \mathrm{Area}(\Omega_m^*)\left(u_{\Omega^*}(p_{\Omega_m^*}) - u_{\Omega^t}(p_{\Omega_m^*})\right)^2$, and normalize the resulting value by that of the initial mesh for comparability across PDEs. This metric captures both the maximum localized errors by punishing outliers, and the overall error across the domain. We evaluate both the mean error and an approximation of the maximum error over the mesh as additional metrics in Appendix D.6. Appendix F.1 lists all further algorithm and neural network hyperparameters.

**Graph Features.** The features $\mathbf{X}_v$ of each node $v \in \mathcal{V}$ consist of the environment timestep, the element area, the distance to the closest boundary, and the mean and standard deviation of the solution on the element's vertices. Edge features $\mathbf{X}_e$ are defined as Euclidean distances between element midpoints. We omit absolute positions to ensure that the observations are equivariant under the Euclidean group [38, 37] to utilize the underlying symmetry of the task. We use additional task-dependent node features for some considered systems of equations, as described in Appendix B.

**Systems of Equations** We experiment on various 2D elliptical PDEs, namely the Laplace equation, the Poisson equation, a Stokes flow task, a linear elasticity example, and a non-stationary heat diffusion equation. The domains are L-shapes, rectangles with a square hole or multiple rhomboid holes, and convex polygons. Figure 2 shows exemplary ASMR refinements on all tasks and briefly explains the challenge associated with each task. The PDEs and the FEM are implemented using *scikit-fem* [89], and we use conforming triangular meshes and linear elements unless mentioned otherwise. The code provides OpenAI gym [40] environments for all tasks. We define the systems of equations and their specific features in Appendix B.

**Baselines.** We adapt several recent RL methods [28, 29, 30] that were originally designed for non-stationary AMR as baselines for our application focusing on stationary refinements. We use our error estimates as the basis of all reward calculations for comparability but otherwise calculate the rewards as described in the respective papers. *Single Agent* [28] predicts a categorical action over the mesh to mark the next element for refinement. *Sweep* [30] trains a single-agent policy by randomly sampling an element on the mesh and deciding its refinement based on local features and a global resource budget. During inference, each timestep consists of a sweep over the full mesh that may mark each element. Finally, *VDGN* [29] estimates a global Q-function as the sum of agent-wise local Q-functions. As the PPO version of *VDGN* has no Q-Function, we decompose the value function as the sum of value functions of the individual elements, yielding a *VDGN*-like baseline in the case of the PPO version. We use an MPN policy for *Single Agent* and *VDGN*, while *Sweep* utilizes a simple MLP. Hyperparameters and further details are provided in Appendix F.2

We also compare to a traditional error-based *Oracle Error Heuristic* [90, 91, 30]. Given a refinement threshold $\theta$, the *Oracle Error Heuristic* iteratively refines all elements $\Omega_i^t$ for which $\mathrm{err}(\Omega_i^t) > \theta \cdot \max_j \mathrm{err}(\Omega_j^t)$. As we are also interested in the reduction of the maximum error, we analogously define the *Maximum Oracle Error Heuristic*, which uses the maximum error per element $\max_{\Omega_m^* \subseteq \Omega_i^t} \left| u_{\Omega^*}(p_{\Omega_m^*}) - u_{\Omega^t}(p_{\Omega_m^*}) \right|$ as a surrogate error estimate. Note that these baselines require the fine-grained reference mesh $\Omega^*$, which is expensive to compute and thus usually unavailable during inference. As a substitute, we consider the commonly used *ZZ Error*, which uses the superconvergent patch recovery process to estimate an error per mesh element [21]. Similar to the *Oracle Error Heuristic*, these estimates are combined with a refinement threshold $\theta$ to iteratively refine the mesh. The *ZZ Error* generally produces smooth error estimates as the recovery process requires averaging over neighboring mesh elements, which can in some cases lead to more coherent refinements when compared to the *Oracle Error Heuristic*. The heuristics act on local element information and greedily refine elements with a high error rather than elements for which a refinement would lead to a high reduction in error. As such, they may select sub-optimal refinements for globally propagating errors, which is a well-known issue for elliptic PDEs [92, 30]. In contrast, RL methods learn to directly maximize the decrease in error, allowing them to find long-term strategies that also take the local receptive fields of the individual agents into account.

**Additional Experiments.** We conduct a series of ablation experiments to determine which parts of ASMR make it uniquely effective. We look at both the area scaling and the spatial decomposition of the reward in Equation 3 and ablate different node features and the number of training PDEs that are used. Additionally, we consider an alternate reward formulation that uses the maximum error per element instead of its average as detailed in Appendix C.2. Due to their importance for practical applications, we further experiment with both generalization to unseen and larger domains and the improvements in runtime for ASMR compared to a uniform refinement.

# 5   Results

**Quantitative Results.** We visualize the mesh quality quantitatively with a Pareto plot of the number of elements and the remaining error. We plot one point per trained policy, which represents the interquartile mean [93] of this policy when evaluated on 100 evaluation environments. We further

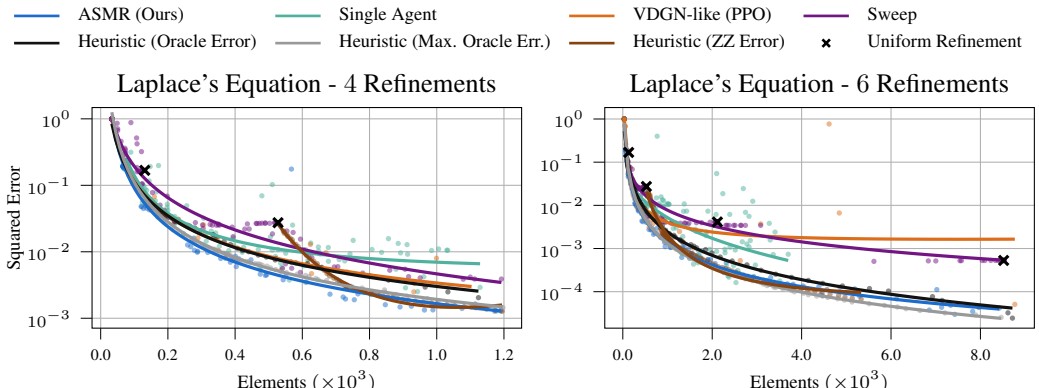

Figure 4: Pareto plot of normalized squared errors and number of final mesh elements for Laplace's Equation. (Left) For only 4 refinement steps, all learned methods perform well and significantly improve upon a uniform refinement. (Right) Scaling to 6 refinement steps, the learned baselines become less stable and in some cases fail to provide refinements that are better than uniform. In contrast, ASMR consistently provides high-quality refinements and is on par with or better than the heuristics in both cases.

provide a log-log quadratic regression over the aggregated results of each method as a general trend-line. To enhance visibility and focus on typical behavior, we exclude sporadic outliers from the baseline methods that produce degenerate meshes with an excessively high number of elements. For all learned methods, we experiment with both PPO and DQN as the RL backbone on the Poisson equation in Appendix D.1. All learned methods, including ASMR yield better results with PPO, indicating that an on-policy algorithm is favorable when dealing with action and observation spaces of varying size. Similarly, we compare the Graph Attention Network (GAT)-like [94] architecture proposed by *VDGN* [29] to MPNs in Appendix D.2, finding that ASMR works well for both architectures, while *VDGN* performs better with MPNs. We consequently use PPO and MPNs in all other experiments. Appendix D.3 compares the *ZZ Error* estimator for different initial refinement levels. As the results show that a sufficiently fine initial mesh is important, we start each refinement procedure for the *ZZ Error Heuristic* with two uniform refinements. We note that this initial tuning prevents coarse refinements and is not needed for our method.

Using these results, Figure 4 compares the different approaches on Laplace's equation. The left side of Figure 4 shows that all methods work in a simple setup on par with experiments from previous work. Here, 4 refinement steps are used for all methods except for *Single Agent*, which instead refines 4 times fewer elements. On the right side, scaling to 6 refinement steps and significantly more elements, only ASMR effectively handles larger instances while learned methods falter. Notably, ASMR also outperforms the *Oracle*, *Maximum Oracle*, and *ZZ Error Heuristics*. These results demonstrate the effectiveness of our Swarm RL framework for learning non-greedy refinement strategies for static meshes. Specifically, ASMR refines elements with a high potential for error reduction over the heuristics' strategy of targeting elements with high error. Figure 5 provides results on the remaining tasks. Appendix D.6 presents additional results using a mean error and a smooth version of a maximum error metric. ASMR clearly outperforms the learned baselines on all tasks and is generally competitive with or better than the *Heuristics*. Both heuristics improve over the RL methods on the Stokes flow task, likely because the task requires high precision for both the inlet and on inner boundaries near regions of high flow velocity.

**Qualitative Results.** Figure 2 shows refinements of ASMR on randomly sampled systems of equations for all considered tasks. The refinement strategy adapts to the given task, providing an efficient trade-off between simulation accuracy and the number of elements used. Figure 3 visualizes the refinements of ASMR on a randomly sampled heat diffusion problem. ASMR refines based on the element penalty $\alpha$, yet always focuses on the heat source and its path. Appendix G.1 provides additional ASMR visualizations for all tasks, and Appendix G.2 visualizes all methods on Poisson's equation to showcase common refinement behaviors. Appendix G.3 presents the iterative marking procedure of our approach on an exemplary Poisson problem.

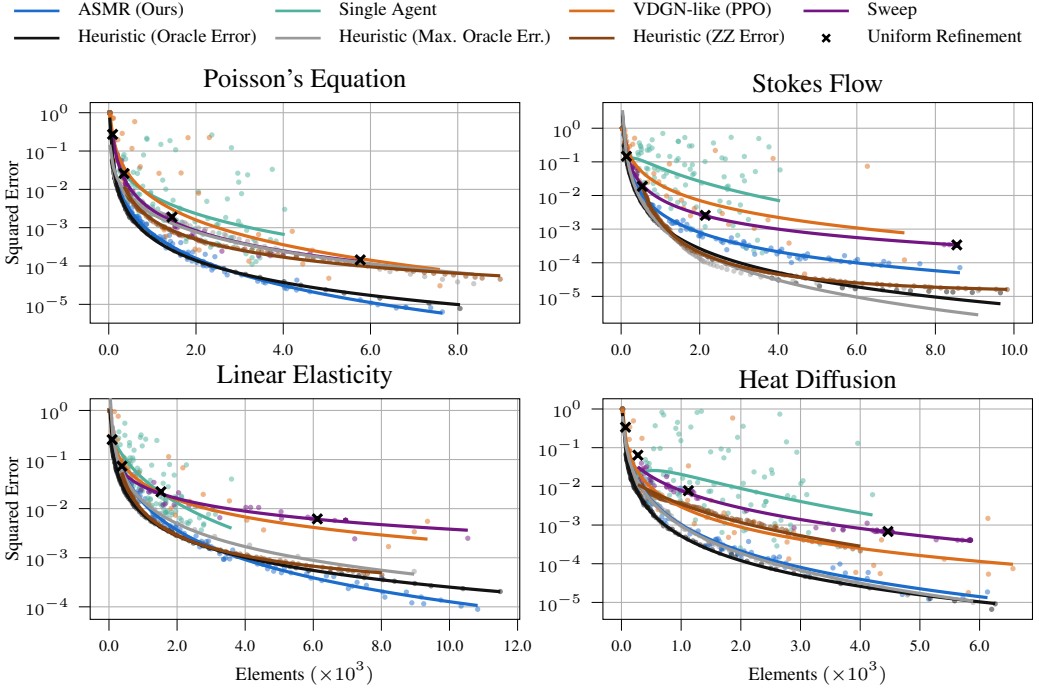

Figure 5: Pareto plot of normalized squared errors and number of final mesh elements across different tasks. All methods generally work well for relatively small instances, but *Single Agent*, *VDGN*-like and *Sweep* break down for larger meshes. Our method uniquely scales to meshes to thousands of elements and consistently outperforms these learned baselines on all tasks and while performing on par with or better than the *Oracle*, *Maximum Oracle* and *ZZ Error Heuristics* in most cases.

**Ablations.** The reward proposed in Equation 3 combines an area scaling per element with a spatial allocation of the decrease in error to the individual mesh elements. Figure 6 investigates these decisions. We find that combining both features is uniquely responsible for the effectiveness of our method, suggesting that the spatial reward's limited expressiveness for small elements is compensated by area scaling. However, the area scaling can only be leveraged if it is allocated to individual mesh elements, as it may introduce excessive reward noise on the full mesh. The maximum reward variant of Appendix C.2 explicitly minimizes the maximum error of the mesh, resulting to refinements of similar quality than those created by ASMR using the reward in Equation 3. We thus use the latter as it simplifies the error estimate in the reward function and better aligns with existing work.

We evaluate the effect of different parameters for the target mesh resolution in Appendix D.4, finding that ASMR provides meshes with considerably more consistent numbers of elements for a given target resolution than the other learned methods. Additional ablations in Appendix D.5 show that 100 training PDEs are sufficient and that adding absolute positions in the node features is detrimental, while providing solution information and load function evaluations improves performance.

**Generalization Capabilities and Runtime Experiments.** Table 1 compares the wall-clock time of ASMR trained on a variant of the Poisson task with that of the reference $\Omega^*$, showing that our method provides a speedup of more than factor 100 compared to computing a uniform mesh for large domains. The evaluation uses load functions with 16 Gaussian modes and spiral-shaped of varying sizes for the same average initial element size. Appendix E.1 provides details for the training environments and the spiral-shaped evaluation domain, as well as results on larger domains and the associated improvement in runtime. These results includes an ASMR visualization of a refinement of a $20 \times 20$ spiral domain with more than $50\,000$ elements. Appendix E.2 additionally shows the exceptional generalization capabilities of ASMR across various domains and load functions for Poisson's equation on $1 \times 1$ domains. Appendix E.3 presents further runtime comparisons for all tasks and shows that ASMR provides a task-dependent speedup of factor 2 to 30 over a uniform refinement.

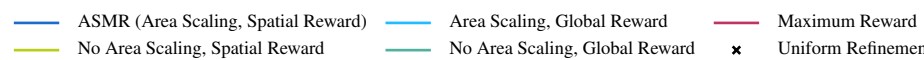

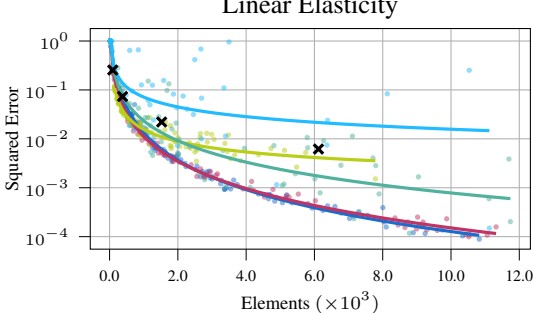

Linear Elasticity

| Domain | Elements | Time[s] | Speedup[×] |
|--------|----------|---------|-----------|
| $2 \times 2$ | 4208 | 0.19 | 14.6 |
| $3 \times 3$ | 4185 | 0.23 | 29.6 |
| $4 \times 4$ | 4391 | 0.28 | 47.1 |
| $5 \times 5$ | 5120 | 0.35 | 67.3 |
| $6 \times 6$ | 5462 | 0.41 | 90.1 |
| $7 \times 7$ | 6515 | 0.51 | 114.8 |
| $8 \times 8$ | 7506 | 0.60 | 131.4 |

Figure 6: Pareto plot of normalized squared errors and number of final mesh elements for the Linear Elasticity task for different reward ablations. ASMR benefits from the area scaling term of Equation 3 and a spatial reward formulation. The maximum reward of Appendix C.2 performs similar to that of Equation 3.

Table 1: Elements and speedup versus uniform refinement for achieving a normalized squared error of 0.001 for different domain sizes on Poisson's Equation. The elements required for the error threshold grow slower than the domain size, indicating fewer areas of significant error in larger domains. Thus, ASMR offers increasing speedups as the size of the domain increases.

The generalization capabilities in combination with the fast runtime of our method offer substantial advantages in practical engineering applications. A policy can be trained on small, cost-effective environments and then deployed on much larger and dynamically changing setups during inference. These generalization traits arise from the MPN architecture and the utilized observation graphs, which both lead to refinement strategies based on local element neighborhoods rather than global meshes.

# 6   Conclusion

We present a novel Adaptive Mesh Refinement method that uses Swarm Reinforcement Learning to iteratively refine meshes for efficient solutions of Partial Differential Equations. Our approach, Adaptive Swarm Mesh Refinement (ASMR), treats each mesh element as an agent and trains all agents under a shared policy using Graph Neural Networks and a novel per-agent reward formulation. ASMR gracefully scales to meshes with thousands of elements without requiring an error estimate during inference. In our experiments focused on static meshes, ASMR demonstrates strong performance in handling complex refinements. The method significantly outperforms both existing Reinforcement Learning-based approaches and traditional refinement strategies, achieving a mesh quality comparable to expensive oracle-based error heuristics.  Once trained, the ASMR policy generalizes well to different forcing functions and significantly larger problem domains. In terms of runtime, our method outperforms uniform refinements by up to 30 times on domains similar in scale to the training set, and by over 100 times in larger evaluation setups.

**Broader Impact** Our proposed Adaptive Mesh Refinement technique can positively impact various fields relying on computational modeling and simulation. By reducing simulation times while maintaining high precision, this technology enables researchers to explore a wider range of scenarios. However, like any powerful tool, there are potential negative impacts, such as the development of advanced weapon models or exploitation of resources.

**Limitations and Future Work** Our approach solves the partial differential equation after each refinement step, which requires a considerable amount of computation time. In future work, we will explore using Swarm RL for refinement strategies from the raw geometry and boundary conditions to further speed up our approach. We currently use relatively simple message passing networks for our policy, and want to optimize the network architecture to include, e.g., long-range message passing. Lastly, this work only considers 2D problems, static meshes with triangular elements, and comparatively simple domains. Here, we want to extend and modify our approach to quadrilateral meshes, time-dependent refinement and coarsening operations, and 3-dimensional domains.

## Acknowledgments and Disclosure of Funding

NF was supported by the BMBF project Davis (Datengetriebene Vernetzung für die ingenieurtechnische Simulation). This work is also part of the DFG AI Resarch Unit 5339 regarding the combination of physics-based simulation with AI-based methodologies for the fast maturation of manufacturing processes. The financial support by German Research Foundation (DFG, Deutsche Forschungsgemeinschaft) is gratefully acknowledged. The authors acknowledge support by the state of Baden-Württemberg through bwHPC, as well as the HoreKa supercomputer funded by the Ministry of Science, Research and the Arts Baden-Württemberg and by the German Federal Ministry of Education and Research.

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

# A Message Passing Network Architecture

Given a graph $\mathcal{G} = (\mathcal{V}, \mathcal{E}, \mathbf{X}_{\mathcal{V}}, \mathbf{X}_{\mathcal{E}})$, Message Passing Networks (MPN) [33, 38, 39] are GNNs consisting of *L Message Passing Steps*. Each step $l$ receives the output of the previous step and updates the features $\mathbf{X}_{\mathcal{V}}$, $\mathbf{X}_{\mathcal{E}}$ for all nodes $v \in \mathcal{V}$ and edges $e \in \mathcal{E}$. Using linear embeddings $\mathbf{x}_v^0$ and $\mathbf{x}_e^0$ of the initial node and edge features, the $l$-th step is given as

$$\mathbf{x}_e^{l+1} = f_{\mathcal{E}}^l(\mathbf{x}_v^l, \mathbf{x}_u^l, \mathbf{x}_e^l), \text{ with } e = (u, v),$$

$$\mathbf{x}_v^{l+1} = f_{\mathcal{V}}^l(\mathbf{x}_v^l, \bigoplus_{e=(v,u)\in\mathcal{E}} \mathbf{x}_e^{l+1}).$$

The operator $\oplus$ is a permutation-invariant aggregation such as a sum, max, or mean operator. Each $f_{\cdot}^l$ is a learned function that we generally parameterize as a simple MLP. The network's final output is a learned representation $\mathbf{x}_v^L$ for each node $v \in \mathcal{V}$.

# B Systems of Equations

In its most general form, the FEM is used to approximate the solution $\boldsymbol{u}(\boldsymbol{x})$ that satisfies the weak formulation $\forall \boldsymbol{x} : \forall \boldsymbol{v}(\boldsymbol{x}) : a(\boldsymbol{u}(\boldsymbol{x}), \boldsymbol{v}(\boldsymbol{x})) = l(\boldsymbol{v}(\boldsymbol{x}))$ of the underlying system of equations for the set of test functions $\boldsymbol{v}$. In the following, we describe the specific equations and boundary conditions used for our experiments.

## B.1 Laplace's Equation

Let $\Omega$ be a domain with an inner boundary $\partial \Omega_0$ and an outer boundary $\partial \Omega_1$. We seek a solution $u(\boldsymbol{x})$ that satisfies the weak formulation of the Laplace Equation

$$\int_{\Omega} \nabla u(\boldsymbol{x}) \cdot \nabla v(\boldsymbol{x}) \, \mathrm{d}\boldsymbol{x} = 0$$

for all test functions $v(\boldsymbol{x})$. Additionally, the solution has to satisfy the Dirichlet boundary conditions

$$u(\boldsymbol{x}) = 0, \boldsymbol{x} \in \partial \Omega_0 \quad \text{and} \quad u(\boldsymbol{x}) = 1, \boldsymbol{x} \in \partial \Omega_1.$$

We use a unit square $(0, 1)^2$ for the outer boundary $\partial \Omega_0$ of the domain and add a randomly sampled square hole, whose borders are considered to be the inner boundary $\partial \Omega_1$. The size of the hole is sampled from the uniform distribution $U(0.05, 0.25)^2$, and its mean position is sampled from $U(0.2, 0.8)^2$. We add the closest distance to the inner boundary as an additional node feature.

## B.2 Poisson's Equation

The weak formulation of the considered Poisson problem is given as

$$\int_{\Omega} \nabla u(\boldsymbol{x}) \cdot \nabla v(\boldsymbol{x}) \, \mathrm{d}\boldsymbol{x} = \int_{\Omega} f(\boldsymbol{x}) v(\boldsymbol{x}) \, \mathrm{d}\boldsymbol{x} \quad \forall v.$$

Here, $f(\boldsymbol{x}) : \Omega \to \mathbb{R}$ denotes the load function and $v(\boldsymbol{x})$ the test function. In addition to the weak formulation, the solution must be zero on the boundary $\partial \Omega$ of the domain $\Omega$. We model Poisson's Equation on L-shaped domains $\Omega$, using a rectangular cutoff whose lower left corner is sampled from $p_0 \sim U(0.2, 0.95)^2$, resulting in a domain $\Omega = (0, 1)^2 \backslash (p_0 \times (1, 1))$. On this domain, we sample a Gaussian Mixture Model with 3 components. Each component's mean is sampled from $U(0.1, 0.9)^2$, and we use rejection sampling to ensure that all means lie within the domain. The components' covariances are determined by first drawing diagonal covariances, where each dimension is drawn independently from a log-uniform distribution $\exp(U(\log(0.0003, 0.003)))$. The diagonal covariances are then rotated by a random angle in $U(0, 180)$ to produce Gaussians with a full covariance matrix. The component weights are drawn from the distribution $\exp(N(0, 1)) + 1$ and subsequently normalized, where the 1 in the end is used to ensure that all components have relevant weight. The evaluation of the load function $f$ at the respective face midpoint is added as a node feature.

## B.3 Stokes flow

Let $\boldsymbol{u}(\boldsymbol{x})$ be the velocity field and $p(\boldsymbol{x})$ the pressure field. We consider a Stokes flow of a fluid through a channel. Therefore, we seek a solution $u$ and $p$, which satisfy the weak formulation of the Stokes flow without a forcing term

$$\nu \int_\Omega \nabla \boldsymbol{v} \cdot \nabla \boldsymbol{u} \, \mathrm{d}\boldsymbol{x} - \int_\Omega (\nabla \cdot \boldsymbol{v}) p \, \mathrm{d}\boldsymbol{x} = 0 \quad \forall \boldsymbol{v}$$

$$\int_\Omega (\nabla \cdot \boldsymbol{u}) q \, \mathrm{d}\boldsymbol{x} = 0 \quad \forall q,$$

wherein $\boldsymbol{v}(\boldsymbol{x})$ and $q(\boldsymbol{x})$ denote the test functions [95]. We define the inlet-profile as

$$\boldsymbol{u}(x = 0, y) = u_\mathrm{P} y(1 - y) + \sin(\varphi + 2\pi y).$$

At the outlet, the gradient of velocity $\nabla \boldsymbol{u}(x = 1, y) = \boldsymbol{0}$ is set to zero. Additionally, we assume a no-slip condition $\boldsymbol{u} = \boldsymbol{0}$ at all boundaries except for the inlet and the outlet. For stability purposes, we use $P_1/P_2$ Taylor-Hood-elements, i.e., quadratic shape functions for the velocity and linear shape functions for the pressure [96]. We sample the quadratic part $u_\mathrm{P}$ of the velocity inlet from a log-uniform distribution $\exp(U(\log(0.5, 2)))$. The class of domains uses a unit square for the outer boundary and 3 rhomboid holes with length $0.4$ and height $0.2$ whose centers are set to $y \in \{0.2, 0.5, 0.8\}$ in y-direction and randomly sampled from $U(0.3, 0.7)$ in x-direction. We optimize the meshes for the prediction accuracy of the velocity in $x$ and $y$ direction and calculate the overall error as the norm of these errors.

## B.4 Linear Elasticity

We are looking for the steady-state deformation of a solid under stress, due to displacements at the boundary of the part $\partial \Omega$. Here, we are interested in both the norm of the deformation and the norm of the stress. The weak formulation of the considered problem on the domain $\Omega$ without body forces is given as [97]

$$\int_\Omega \boldsymbol{\sigma}(\boldsymbol{\varepsilon}(\boldsymbol{u})) : \boldsymbol{\varepsilon}(\boldsymbol{v}) \, \mathrm{d}\boldsymbol{x} = 0.$$

Here, $\boldsymbol{u}(\boldsymbol{x})$ is the displacement field, $\boldsymbol{v}(\boldsymbol{x})$ is the test function, and $\boldsymbol{\varepsilon}(\boldsymbol{u}) = \frac{1}{2}(\nabla \boldsymbol{u} + (\nabla \boldsymbol{u})^\top)$ is the strain tensor. $\boldsymbol{\sigma}(\boldsymbol{\varepsilon})$ is the stress tensor, which is given as $\boldsymbol{\sigma}(\boldsymbol{\varepsilon}) = 2\mu\boldsymbol{\varepsilon} + \lambda\mathrm{tr}(\boldsymbol{\varepsilon})\boldsymbol{I}$ in a linear-elastic and isotropic case. The Lamé parameters $\lambda = \frac{E\nu}{(1+\nu)(1-2\nu)}$ and $\mu = \frac{E}{2(1+\nu)}$ can be calculated with the problem specific Young's modulus $E = 1$ and the Poisson ratio $\nu = 0.3$. The displacement $\boldsymbol{u}(x = 0, y) = \boldsymbol{u}_0$ on the left side of the boundary is specified by a task-dependent parameter $\boldsymbol{u}_0$, whereas the displacement $\boldsymbol{u}(x = L, y) = 0$ is set to zero on the right boundary. The stress $\boldsymbol{\sigma} \cdot \boldsymbol{n} = \boldsymbol{0}$ is zero normal to the boundary at both the top and bottom of the part. We use the same class of L-shaped domains as in the Poisson problem in Section B.2 and set $u_P$ by drawing a random angle from $U[0, \pi]$ to pull on the domain from different angles, and add random magnitude from $U(0.2, 0.8)$. We add the task-dependent displacement $u_P$ as a feature to all nodes. We are interested in the norm of the displacement field $u$ and the resulting Von-Mises stress, giving us a 2-dimensional objective. We weight both dimensions equally in the reward.

## B.5 Non-stationary Heat Diffusion

We consider a non-stationary thermal diffusion problem defined by the weak formulation

$$\int_\Omega \frac{\partial u}{\partial t} \, \mathrm{d}\boldsymbol{x} + \int_\Omega a\nabla u \cdot \nabla v \, \mathrm{d}\boldsymbol{x} = \int_\Omega fv \, \mathrm{d}\boldsymbol{x} \quad \forall v,$$

wherein $u$ denotes the temperature, $v$ the test function, $a$ the thermal diffusivity and $f$ a heat distribution, given as

$$f = q \exp\left(-100\left((x - x_p(\tau)) + (y - y_p(\tau))\right)\right).$$

The position of the maximum heat entry $\boldsymbol{p}_\tau(\tau) = (x_p(\tau), y_p(\tau))$ is changing over time, while its magnitude is scaled by a factor $q$. The temperature $u \in \partial\Omega$ is set to zero on all boundaries. For the

time-integration, the implicit Euler method is applied. We use a total of $\tau_{\max} = 20$ time steps in $\{0.5, \ldots, 10\}$, a scaling factor of $q = 1000$ and a diffusivity $a = 0.001$. The position of the heat source at step $\tau$ is linearly interpolated as $\boldsymbol{p}_\tau = \boldsymbol{p}_0 + \frac{\tau}{\tau_{\max}}(\boldsymbol{p}_{\tau_{\max}} - \boldsymbol{p}_0)$, where the start and goal positions $\boldsymbol{p}_0$ and $\boldsymbol{p}_{\tau_{\max}}$ are randomly drawn from the domain. To create our domains, we start with 10 points that are equidistantly placed on a circle with center $(0.5, 0.5)$ and radius $0.4$. Each point is distorted by a random value drawn from $U(-0.2, 0.2)^2$. We then normalize the resulting points to be in $(0, 1)^2$ and calculate the convex hull. The result is a family of convex polygons with up to 10 vertices. We measure the error and solution of the final simulation step, and provide the distance to the start and end position of the heat source as additional node features for each element.

## C  Further Experiments

### C.1  Experiment Details

All experiments are repeated for 10 random seeds with randomized PDEs and network parameters. All domains are normalized to be in $(0, 1)^2$ unless mentioned otherwise. The initial meshes are created using meshpy[2]. For practical purposes, we add an element threshold $\beta_{\max}$ in our environments, and terminate an episode with a large negative reward when this threshold is exceeded. We train all policies on 100 training PDEs and evaluate the resulting final policies on 100 different evaluation PDEs that we keep consistent across random seeds for better comparability. All experiments are run for up to 2 days on 8 cores of an Intel Xeon Platinum 8358 CPU. In terms of total compute, we train 4 different learned methods, namely the 3 RL baselines and our method, on 5 separate tasks. Each experiment is repeated for 10 different target mesh resolutions and 10 repetitions, resulting in $5 \cdot 4 \cdot 10 \cdot 10 = 2000$ main experiments. Additionally, we use a similar amount of compute for the combined ablations, preliminary experiments and heuristics.

### C.2  Maximum Reward

Equation 3 scales the reduction in error of each element by its area. This modification encourages the policy to focus on smaller elements, effectively shifting the objective from an reduction in average error across the mesh to a minimization of error densities. An alternate way to phrase this objective is to make the reward depend on the reduction in maximum error per element. For this, we modify the error estimate per element of Equation 2 to read

$$\hat{\text{err}}(\Omega_i^t) \approx \max_{\Omega_m^* \subseteq \Omega_i^t} \left| u_{\Omega^*}(p_{\Omega_m^*}) - u_{\Omega^t}(p_{\Omega_m^*}) \right|,$$

and subsequently drop the area scaling and replace the sum in Equation 3 with a maximum, i.e.,

$$\mathbf{r}'(\Omega_i^t) := \left( \text{err}(\Omega_i^t) - \max_j \mathbf{M}_{ij}^t \text{err}(\Omega_j^{t+1}) \right) - \alpha \left( \sum_j \mathbf{M}_{ij}^t - 1 \right).$$

While conceptually simpler than our reward formulation, evaluating the decrease in maximum error only optimizes this objective, which may result in worse meshes when looking at, e.g., the mean error. The left side of Figure 6 compares this alternate reward formulation to that of ASMR. We find that both reward schemes perform similarly. Therefore, we opt for the reward function defined in Equation 3 for simplicity and easier comparison with the baseline methods.

## D  Extended Results

### D.1  Proximal Policy Optimization and Deep Q-Networks.

The left side of Figure 7 shows results on Poisson's Equation for PPO and DQN as the RL backbone for all learned methods. We find that PPO outperforms DQN, suggesting that an on-policy objective is favorable for the changing observation and action spaces of AMR. We use a mean instead of a sum for the agent mapping of the targets of the $Q$-values for the DQN experiments with ASMR as this

---

[2] https://github.com/inducer/meshpy

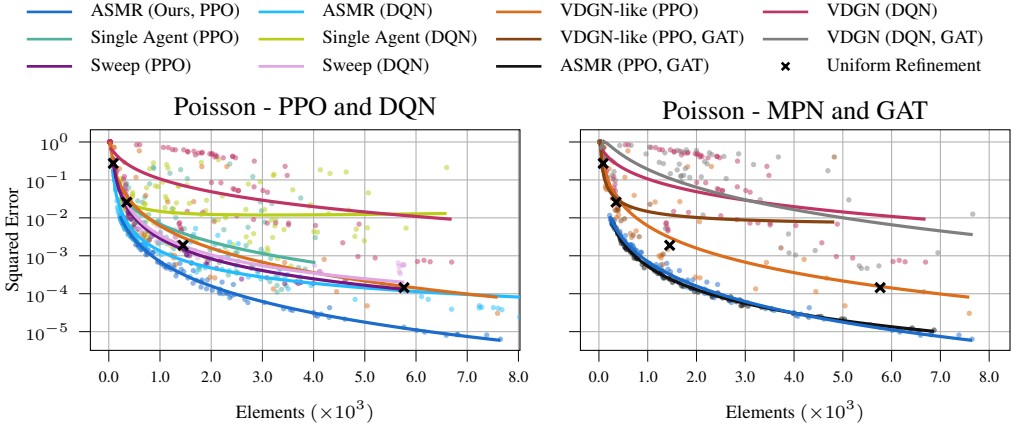

Figure 7: Pareto plot of normalized squared errors and number of final mesh elements on Poisson's equation for (Left) PPO and DQN for all RL baselines and (Right) GAT and MPN for ASMR and *VDGN*(-like). PPO generally results in better performance. ASMR works well for both MPNs and GATs, while the *VDGN*-like baseline is more stable when using MPNs.

seems to experimentally increase training stability. For the *VDGN*-like variant that uses PPO, we factorize the value function instead of the Q-function, i.e., we define the value function of the full mesh as the sum of value functions of the individual mesh elements. We choose PPO for all other experiments as it leads to better performance for all methods.

## D.2 Message Passing and Graph Attention Networks.

The right side of Figure 7 compares MPNs and GATs for ASMR and *VDGN*. For ASMR, the performance between MPNs and GATs is comparable, while *VDGN* seems to be more stable and produce better refinements when using MPNs. We use MPNs for the other experiments as it seems to benefit *VDGN* while decreasing the performance of our method.

## D.3 Initial Meshes for the ZZ Error.

Figure 8 compares the *ZZ Error Heuristic* when directly applied to the initial mesh to variants that instead start each refinement procedure by uniformly refining either once or twice. We find that the method greatly benefits from two initial uniform refinements, likely because the heuristic may not detect gradients for interesting parts of the domain if the corresponding elements are too coarse. Given these results, we use the twice refined version for all experiments, noting that the RL based methods avoid having to tune the initial mesh by design.

## D.4 Target Mesh Resolutions

All RL methods use some parameter to control the number of target elements of the final refined mesh. ASMR and *VDGN* use an element penalty $\alpha$, *Sweep* uses a budget $N_{\max}$, and *Single Agent* different numbers of rollout steps $T$. We visualize evaluations for different target resolutions in Figure 9. The results indicate that ASMR provides meshes with consistent numbers of elements for a given target resolution, while the other RL methods produce meshes with inconsistent numbers of elements over target resolutions. The concrete target resolution parameters for all experiments are found in Table 2.

## D.5 Ablations.

**Node Features.** ASMR utilizes both task-dependent information, such as the evaluation of the load function for Poisson's Equation, and the local solution $u(x)$ per mesh element as part of its observation graph. Here, we experiment how the performance is affected if either of these features is left out. Additionally, we consider a variant where we include explicit $(x, y)$ positions of each element midpoint as node features. The results are shown on the left of Figure 10. We find that

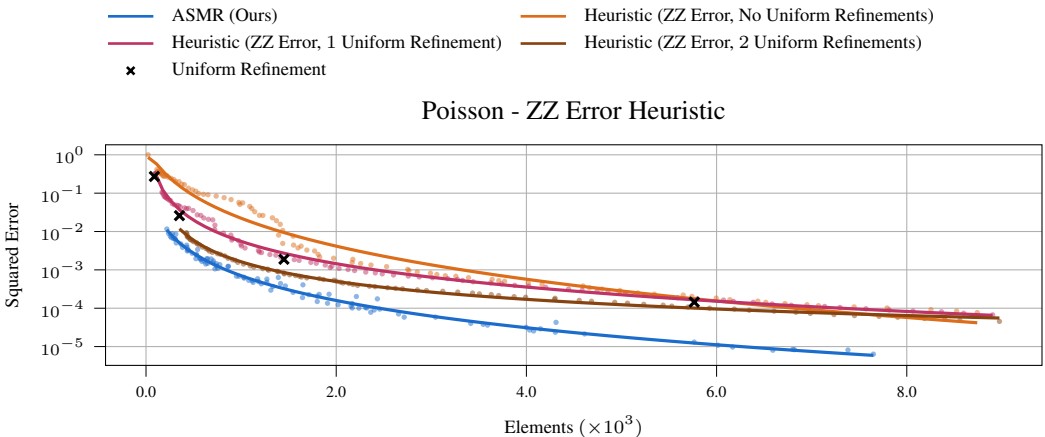

Figure 8: Pareto plot of normalized squared errors and number of final mesh elements on Poisson's equation for the Zienkiewicz-Zhu Error Estimator (*ZZ Error*) *Heuristic* when using either no, 1, or 2 initial uniform refinements. The *ZZ Error Heuristic* produces better refinements when provided with a finer initial mesh at the cost of not being able to produce meshes with very few elements.

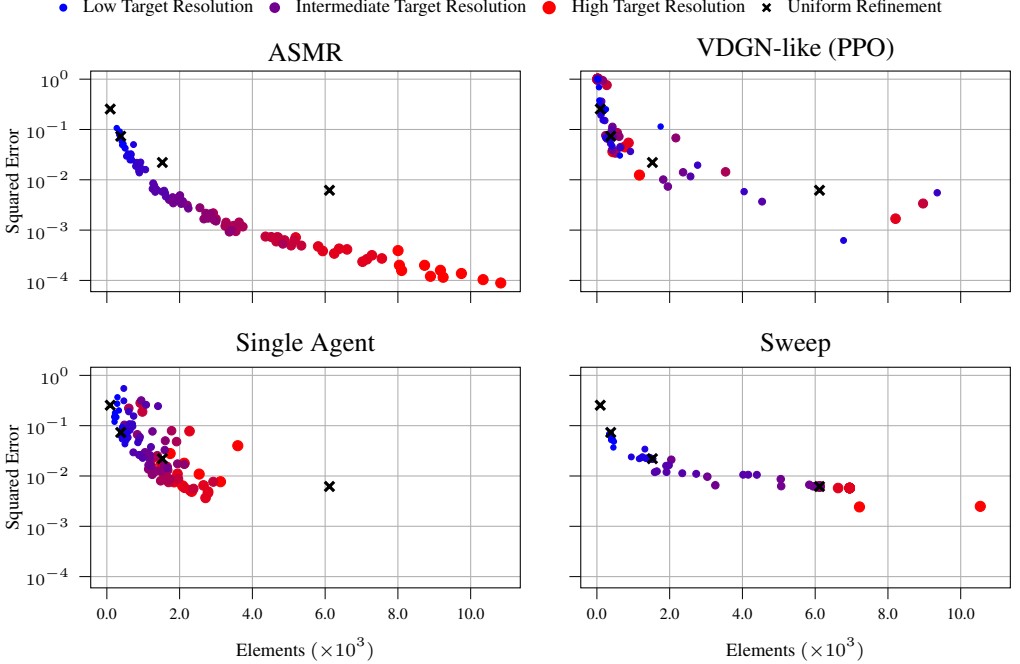

Figure 9: Pareto plot of normalized squared errors and number of final mesh elements for the linear elasticity task for all RL methods. Small blue dots indicate a policy trained on a coarse target mesh resolution, which corresponds to large element penalties $\alpha$ for ASMR and *VDGN*-like, a small budget $N_{max}$ for *Sweep* and a low number of rollout steps $T$ for *Single Agent*. Large red dots correspond to a finer target meshes, and the medium-sized purple dots interpolate between the two. Details on the target resolution parameters are found in Table 2. We find that ASMR provides high-quality refinements with consistent numbers of final mesh elements for any given target resolution, whereas the other methods yield poor-quality refinements (*Sweep*) or inconsistent results (*VDGN*-like, *Single Agent*) for similar target resolutions and different random seeds.

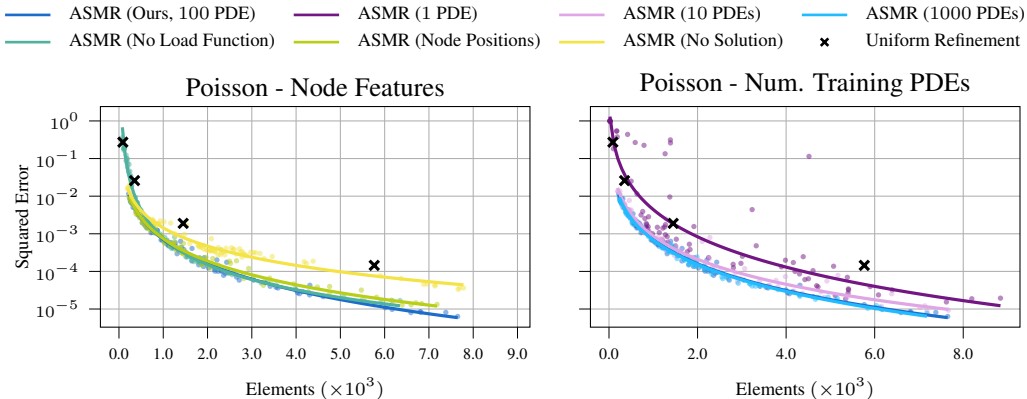

Figure 10: Pareto plot of normalized Top $0.1\%$ of errors and number of final mesh elements for Poisson's Equation. (Left) Omitting either the solution or evaluation of the load function per element leads to a decrease in refinement quality. Explicit node positions slightly decreases performance, likely because they cause the observation to lose equivariance w.r.t., e.g., reflection. (Right) Performance is reduced for fewer training PDEs, but stabilizes around 100 PDEs. Interestingly, ASMR achieves better-than performance when using a single training PDE, meaning that it can generalize from a single training example. This ability is likely a result of our spatial problem formulation.

both the task-dependent features and the solution are important for the performance of our approach. Omitting positional features slightly improves performance, presumably because the features assign a fixed position to each mesh element, causing the observation graph to no longer be equivariant to rotation, translation and reflection. Interestingly, ASMR provides reasonable refinements even without solution information, suggesting that the RL algorithm is able to detect relevant regions of the PDE from just an encoding of the domain and the boundary conditions and forcing functions.

**Number of Training PDEs** Since calculating the fine-grained reference $\Omega^*$ is slow for large meshes and complex tasks, we want to minimize the number of unique PDEs that we need during training. We use 100 PDEs in our other experiments, and additionally visualize results for 1, 10 and 1000 training PDEs on the right of Figure 10. We find that fewer than 100 PDEs lead to less stable and reliable results, and that there is only a minor advantage in using 1000 PDEs compared to our 100. Noticeably, a single training PDE results in suitable refinements, which hints at significant generalization capabilities that are likely granted by our spatial treatment of the underlying task.

### D.6 Alternate Error Metrics

Section 5 evaluates all approaches on the normalized squared error of the mesh. This metric captures both the average error across the domain, leading to a low mean error, and outliers, thus punishing a high maximum error. Here, we additionally present normalized mean and maximum error metrics to provide a more thorough nuanced evaluation. The first directly quantifies the average absolute error of the mesh, which makes it easy to interpret and less sensitive to outliers. The maximum error metric measures the worst-case performance, which is crucial for applications where a single high-error prediction could be costly. Since the maximum remaining error is susceptible to outliers, we approximate it as the average of the Top $0.1\%$ of errors of all integration points $p_{\Omega_m^*}$. For comparability across PDEs, we normalize both metrics by the respective error of the initial mesh $\Omega^0$.

Figure 11 displays the results for all tasks and both alternate metrics. The general trends for both metrics are consistent with that of the squared error in Section 5, with ASMR outperforming all learned baselines while being on par with or better than the *Heuristics* in most cases. The *Oracle Error Heuristic* performs particularly well on the mean error metric, as it selects elements with high integrated error for refinement. Conversely, the *Maximum Oracle Error Heuristic* excels on the top $0.1\%$ error metric, as it specifically targets elements with a high maximum error. Notably, the mean error metric tends to favor more uniform meshes due to its lower sensitivity to outliers, enabling baselines like *Sweep*, which generally produce relatively uniform meshes, to yield better performance here when compared to the other metrics.

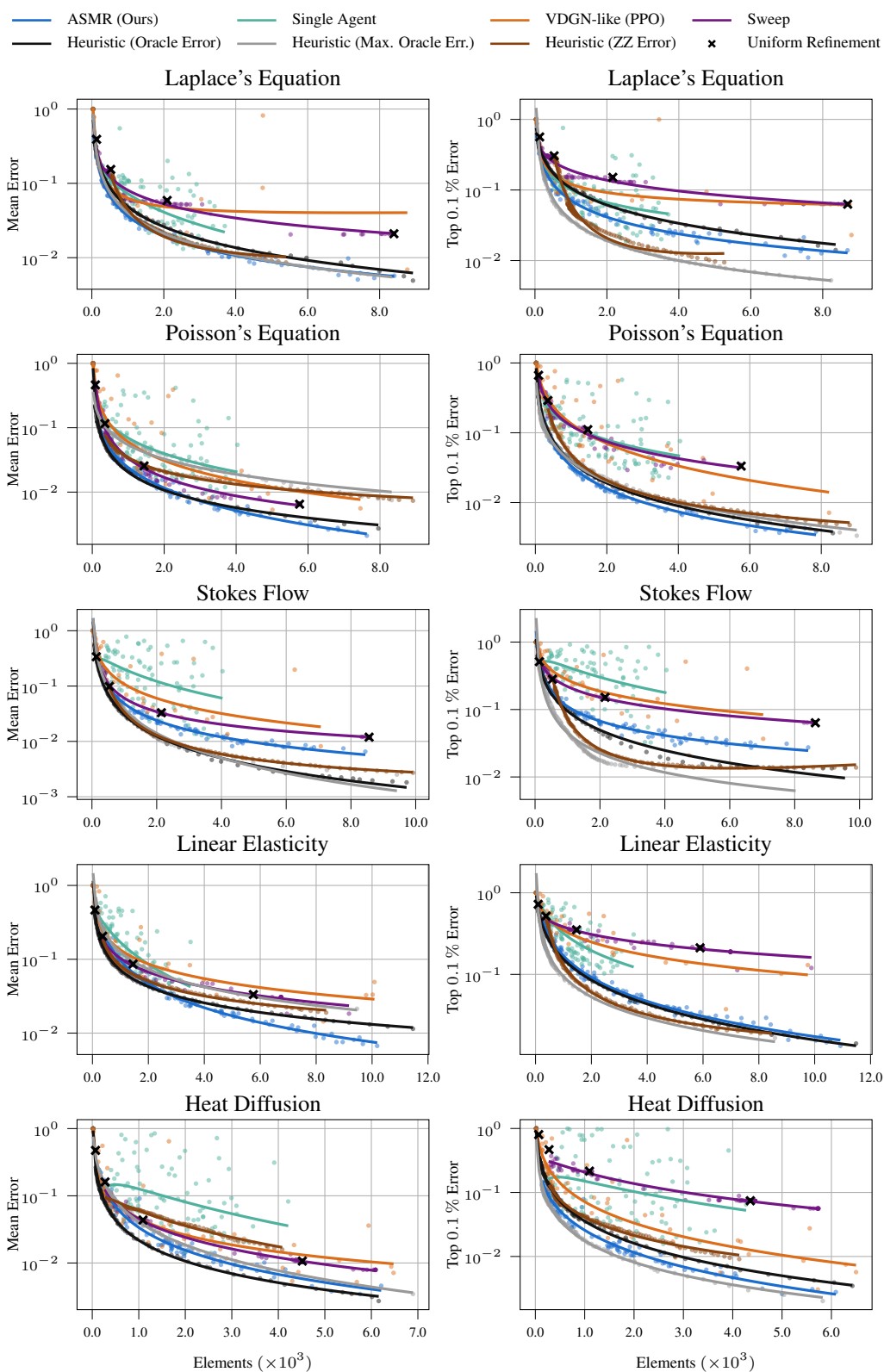

Figure 11: Pareto plot of (left) normalized mean errors and (right) normalized top $0.1\%$ errors compared to number of final mesh elements across different tasks. Performance on both metrics is highly correlated and generally consistent with that of the squared error in Figures 4 and 5. ASMR outperforms all learned baselines on both metrics and all tasks.

# E  Generalization Capabilities and Runtime Experiments

## E.1  Domain Size Generalization

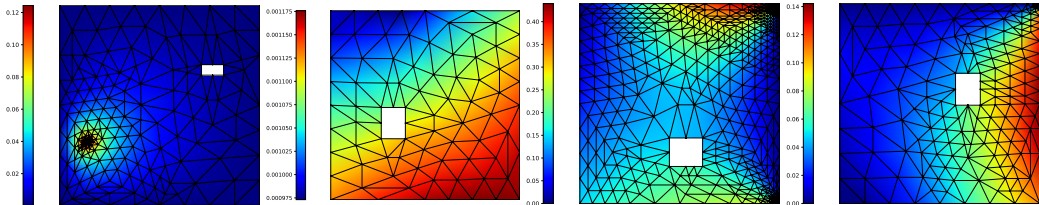

Figure 12: Random training PDEs and ASMR refinements for the domain size generalization experiments. The solutions are on different scales as indicated by the colorbars on the left of the meshes.

We experiment with the abilities of ASMR to generalize to larger domains during inference for Poisson's equation. Such generalization is non-trivial due to the varying boundary conditions and complexities arising from domain scaling, yet extremely useful in practical scenarios where a policy is trained on small and relatively cheap training domains, and then applied to a much larger setups during inference. To generalize to larger domains, we modify the training PDEs to mimic larger mesh segments by altering boundary conditions and load functions. The means of the load function are sampled from a centered unit Gaussian, allowing components outside the mesh. We use domains with random holes for varied initial meshes and apply random Gaussian loads to selected boundary parts as 'inlets'. Examplary training PDEs and ASMR refinements can be seen in Figure 12. We further add an L2 norm of $3e-4$ to combat overfitting and omit the per-domain normalization in favor of a constant normalization factor of $100$, i.e., use $100 \cdot \hat{err}(\Omega)$ instead of $err(\Omega)$ in Equation 3. These modifications can be seen as data augmentation and only affect the training environments without changing the ASMR algorithm. We evaluate the resulting policy on larger, spiral-shaped domains with initial elements of the same size as the evaluation domains. Figure 13 shows that ASMR consistently provides high-quality refinements as the domain increases size (left) while leading to more and more significant speedups when compared to the reference uniform refinement (right). Here, we use a spiral-shaped domain and load functions with 16 randomly placed components. A slice of these figures for a normalized squared error of $0.001$ is provided in the main paper in Table 1.

Figure 14 shows how the same procedure scales to inference on a spiral mesh of size $20 \times 20$ with a load function with $81$ components. The refined mesh has more than $50\,000$ elements, which is several times larger than any refinement shown by previous work. Creating and solving this mesh using ASMR is roughly $100$ times faster than solving the fine-grained reference $\Omega^*$. A close-up for the marked region is shown on the right side of Figure 15. The left side of Figure 15 compares ASMR trained on the generalization environments with the setup used throughout the paper, showing that the additional generalization capabilities only lead to a marginal decrease in performance on the original evaluation PDEs.

## E.2  Same-scale Generalization Capabilities

We additionally visualize ASMR on Poisson's equation on domains of size $1 \times 1$, i.e., of the same size that is seen during training. Here, we utilize the regular training environments without the above augmentation and perform inference on 3 different domain types used in throughout the main experiments, plus a simple rectangular domain $\Omega = (0,1)^2$ and randomly generated trapezoids. We sample 3 random domains per class, and use Gaussian Mixture Model load functions with 1, 3 and 5 components respectively. Figure 16 shows refinements of an ASMR policy with $\alpha = 0.0075$ for the resulting $3 \times 5$ problems. We find that ASMR generalizes across domains and load functions, which is likely a result of the Swarm RL setting, where each mesh element is governed by its own agent.

## E.3  Runtime Comparison.

Finally, we compare the wallclock-time of our approach with that of directly computing the fine-grained uniform mesh $\Omega^*$ on the evaluation PDEs. For ASMR, we measure the cumulative time of

# Poisson - Domain Size Generalization

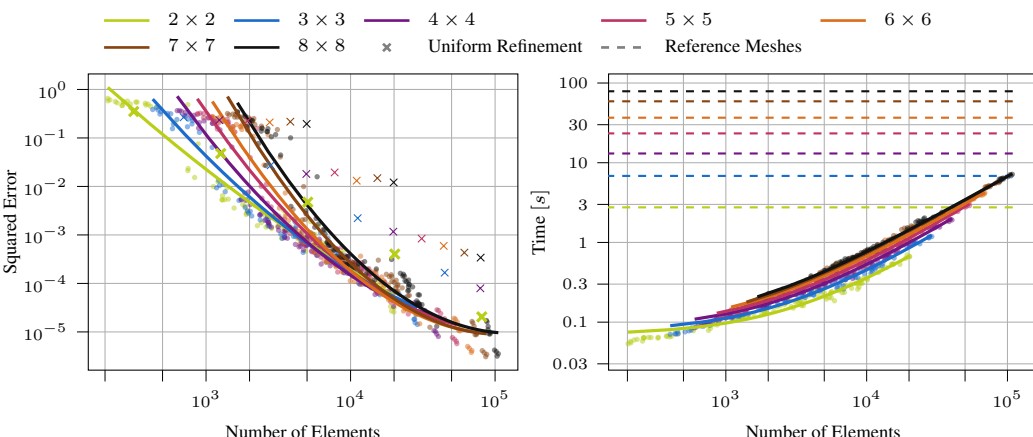

Figure 13: (Left) Pareto plot of the normalized squared error for ASMR evaluated on domains of size $N \times N$. While the initial number of elements increases with the size of the domain, fewer relative elements are needed to achieve a given error threshold, suggesting that there are fewer elements with a significant error in larger domains. (Right) Wallclock-time in seconds of ASMR (points, solid lines) for different numbers of elements compared to the uniform reference $\Omega^*$ (dashed lines). ASMR provides high-quality refinements across different domain sizes, achieving larger and larger speedups when compared to the uniform mesh as the size of the domain increases.

creating an initial coarse mesh, iteratively solving the problem on this mesh, computing the resulting observation graphs after every step, feeding each observation graph to the policy to obtain a set of actions, and using each set of actions to refine the mesh a total of $T = 6$ times. For the uniform mesh, we simply measure the time it takes to refine the coarse mesh 6 times and to subsequently solve the problem on the resulting mesh. We use a single 8-Core AMD Ryzen 7 3700X Processor for all measurements. Figure 17 shows the results for all tasks. We find that our approach is always significantly faster than computing the fine-grained mesh despite the comparatively large computational overhead. Further, the final resolution of the refined mesh produced by our method trades off the wallclock-time of the method, meaning that ASMR can be trained to generate coarser or finer meshes depending on task-specific computational budgets. Notably, for the Stokes flow equations, which use $P_1/P_2$ Taylor-Hood-elements, our method is more than 30 times faster than $\Omega^*$ even for highly refined final meshes. Since the Local Oracle baseline requires the calculation of $\Omega^*$ and otherwise follows a similar iterative refinement procedure, its runtime is dominated by $\Omega^*$.

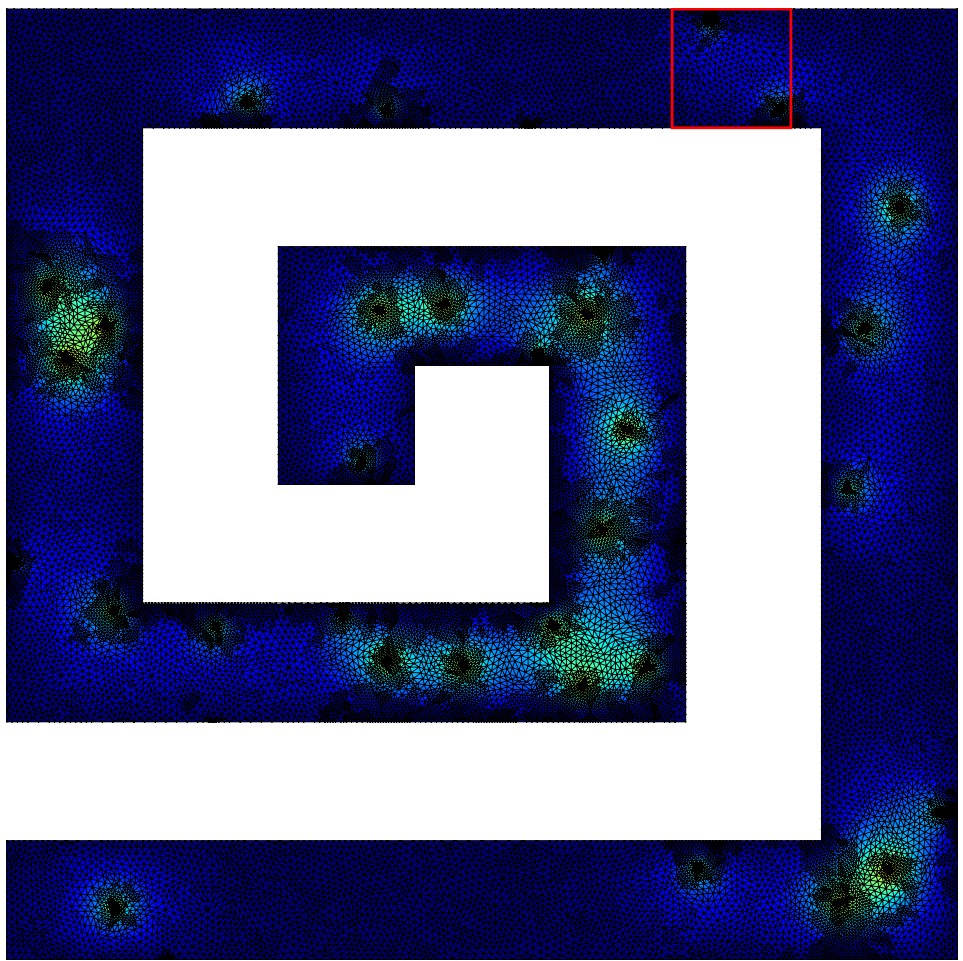

Figure 14: Visualization of a refinement produced by the policy of Figure 12 on a $20 \times 20$ spiral domain. The presented mesh has $53\,189$ elements and ASMR produces it in about 10 seconds on a regular CPU, whereas a uniform refinement $\Omega^*$ takes more than 20 minutes. A close-up of the marked region is shown on the right side of Figure 15. ASMR provides highly accurate refinements for domains that are significantly larger than those seen during training.

# F   Hyperparameters

## F.1   General Hyperparameters

We use the same hyperparameters across all methods and environments unless mentioned otherwise.

**PPO.** We largely follow the suggestions of [98] for our PPO parameters. We train each PPO policy for a total of $400$ iterations. In each iteration, the algorithm samples $256$ environment transitions and then trains on them for $5$ epochs with a batch size of $32$. The value function loss is multiplied with a factor of $0.5$ and we clip the gradient norm to $0.5$. The policy and value function clip ranges are chosen to be $0.2$. We normalize the observations with a running mean and standard deviation. The discount factor is $\gamma = 0.99$ and advantages are estimated via Generalized Advantage Estimate [99] with $\lambda = 0.95$. We compute an agent's advantage by subtracting the agent-wise value estimates from the combination of local and global returns in Equation 4.

**DQN.** For DQN-based approaches, we instead train for $24 * 400 = 9600$ steps, where each step consists of executing an environment transition and then drawing a batch of $32$ samples from the replay buffer for a single gradient update. We additionally draw $500$ initial random replay buffer samples before the first training step. We keep $10000$ transitions in the replay buffer, since each transition represents a full mesh and an action on each graph element. We experimented with both larger replay

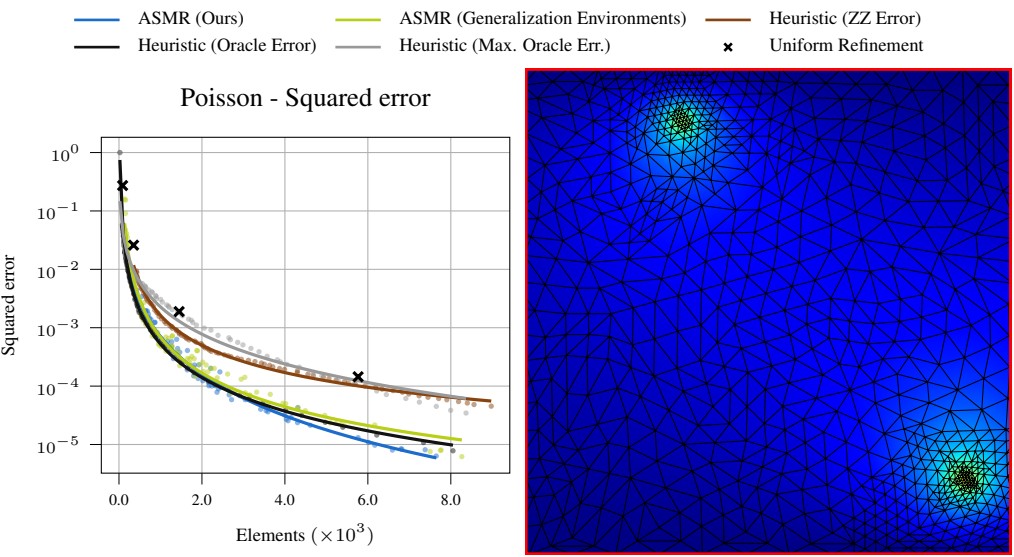

Figure 15: (Left) Pareto plot of the normalized squared error for ASMR trained on regular (blue) and generalizing (green) environments evaluated on $1 \times 1$ evaluation PDEs. (Right) A local region of the mesh in Figure 14. Adding data augmentation in the training environments allows generalization to significantly larger meshes during inference at the cost of slightly decreased refinement quality.

buffers and more training steps in preliminary experiments, finding that neither significantly improve performance, but may lead to very long runtimes and large memory requirements. During training, we draw actions using a Boltzmann distribution over the predicted Q-values per agent, where we linearly decrease the temperature of the distribution from $1$ to $0.01$ in the first $4800$ steps. We find that this action selection strategy leads to more correlated actions when compared to an epsilon greedy action sampling, which stabilizes the training for our iterative mesh refinement problems. We update the target networks using Polyak averaging at a rate of $0.99$ per step. Further, we follow previous work [100] and include a number of common improvements for DQNs in our implementation. These are double Q-learning [101], dueling Q-networks [102] and prioritized experience replay [103].

**Neural Networks.** All networks are implemented in PyTorch [104] and trained using the ADAM optimizer [105] with a learning rate of $3.0e$-$4$ unless mentioned otherwise. All MLPs use $2$ hidden layers and a latent dimension of $64$. We use separate MPNs for the policy and the value function. Each MPN consists of $2$ message passing steps, where each update function is represented as an MLP with *LeakyReLU* activation functions. The policy and value function heads are additional MLPs with *tanh* activation functions acting on the final latent node features of the MPN. All message aggregations $\bigoplus$ are mean aggregations. Additionally, we apply Layer Normalization [106] and Residual Connections [107] independently for the node and edge features after each message passing step.

## F.2 Baseline-Specific Parameters

For *Single Agent*, we use a maximum refinement depth of $10$ refinements per element to avoid numerical instabilities during simulation, skipping actions that try to refine elements that have been refined too often. We consider environment sequences of up to $T = 400$ steps since the method marks only one element at a time. For *Sweep*, the agent is placed on a random mesh element for each training step and may decide not to refine this element, resulting in no change in the mesh. Here, we follow the proposed hyperparameters for this approach and train each rollout for $200$ steps. As this approach is based on purely local agents, we adapt our input features per element to consist of our regular node features, the global resource budget proposed by the authors, the mean solution and area of the element's neighbors and the average distance to them. The global budget is controlled via a maximum number of elements $N_{\max}$, allowing to get refinements of different granularity. To accommodate for less overall changes in the mesh, we increase the number of environment transitions

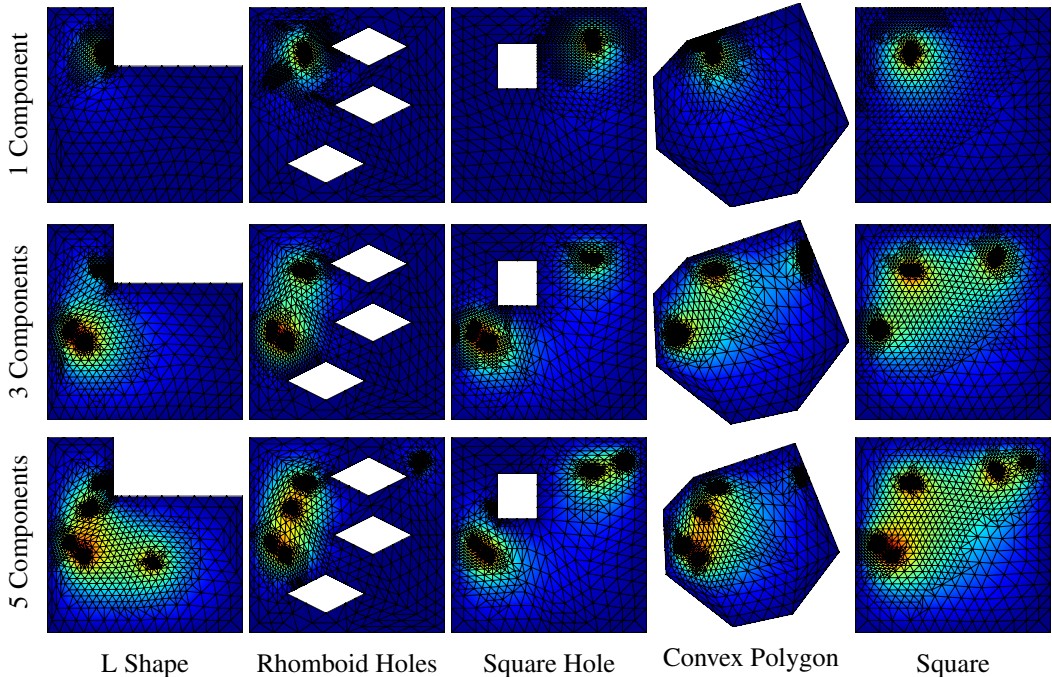

Figure 16: ASMR refinements for different domains and Poisson's equation with a Gaussian Mixture Model load with 1, 3 and 5 components. Even though the ASMR policy is only trained on L-shaped domains with 3 components in the load function and no data augmentation, it generalizes to different domains and loads.

of PPO to 512, and the number of DQN steps to $96 * 400 = 38400$. Finally, we use a learning rate of 1.0e-5 instead of 3.0e-4 for the DQN variant of *VDGN* to stabilize its training.

### F.3 Refinement Hyperparameters

The AMR methods considered in this work use different parameters to control the granularity of the final refined mesh. ASMR and *VDGN* use an element penalty $\alpha$, while *Sweep* considers an element budget $N_{\max}$. *Single Agent* varies the number of rollout steps $T$. For each learned method and task, we choose 10 different values for the refinement parameter that showcase a wide range of final mesh resolutions. For the *Oracle*, *Maximum Oracle*, and *ZZ Error Heuristics* we instead cover a range of up to 100 parameter thresholds $\theta$, yielding one aggregated evaluation result per threshold.

Table 2 lists the different ranges for these parameters for the different tasks. For stability purposes, we set a maximum number of 20 000 elements during training for all experiments except for the *Sweep* baseline, as this baseline uses its own element budget instead. If this number is surpassed, a constant penalty of 1000 is subtracted from the reward and the episode terminates early.

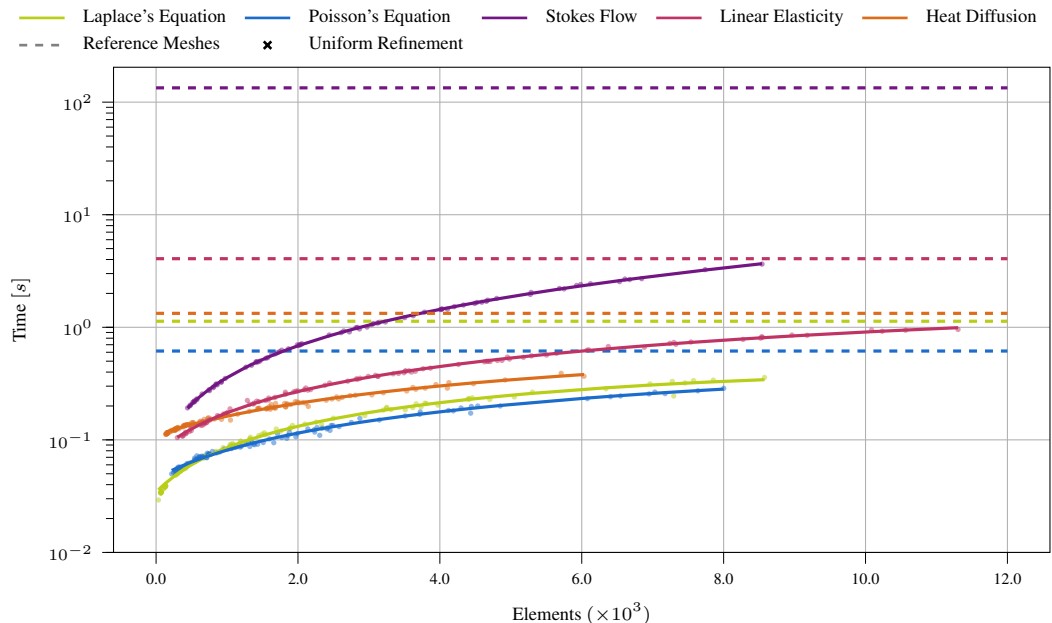

Figure 17: Wallclock-time in seconds of ASMR (points, solid lines) for different numbers of elements compared to the uniform reference $\Omega^*$ (dashed lines). Each task is denoted by its color, and the full lines represent a quadratic regression of the wallclock-time of our method for different numbers of final elements. On average, $\Omega^*$ contains about $10^5$ elements, with concrete numbers varying depending on the domain. Our approach is significantly faster than the reference for all tasks, achieving speedups of factor 2-30 for reasonable numbers of elements.

Table 2: Ranges for the different refinement hyperparameters for all tasks. ASMR and *VDGN* apply an element penalty $\alpha$, but only ASMR scales the area of each element with its area in Equation 3. *Sweep* uses an element budget $N_{\max}$. *Single Agent* varies the number of rollout steps $T$, and the *Oracle Error*, *Maximum Oracle Error* and *ZZ Error Heuristics* use of different error thresholds $\theta$.

| Method | Task | | | | |
|---|---|---|---|---|---|
| | Laplace | Poisson | Stokes Flow | Lin. Elast. | Heat Diff. |
| ASMR ($\alpha$) | $[0.01, 0.3]$ | $[0.002, 0.1]$ | $[0.006, 0.15]$ | $[0.01, 0.15]$ | $[0.003, 0.3]$ |
| *VDGN*(-like) ($\alpha$) | $[2e{-}5, 5e{-}2]$ | $[2e{-}5, 5e{-}2]$ | $[3e{-}4, 5e{-}3]$ | $[1e{-}5, 1e{-}2]$ | $[5e{-}6, 5e{-}3]$ |
| *Sweep* ($N_{\max}$) | $[200, 3000]$ | $[200, 3000]$ | $[200, 3500]$ | $[500, 6000]$ | $[400, 5000]$ |
| *Single Agent* ($T$) | $[25, 400]$ | $[25, 400]$ | $[25, 400]$ | $[25, 400]$ | $[25, 400]$ |
| *Oracle Error* ($\theta$) | $[0.25, 1.00]$ | $[0.1, 1.0]$ | $[0.16, 1.0]$ | $[0.02, 1.0]$ | $[0.03, 1.0]$ |
| *Max. Oracle Err.* ($\theta$) | $[0.20, 1.0]$ | $[0.2, 1.0]$ | $[0.1, 1.0]$ | $[0.01, 1.0]$ | $[0.02, 1.0]$ |
| *ZZ Error* ($\theta$) | $[0.001, 1]$ | $[0.002, 1]$ | $[0.001, 1]$ | $[0.001, 1]$ | $[0.001, 1]$ |

# G   Visualizations

We provide additional visualizations for our method on all tasks, and for all methods on the Poisson task. All visualizations show the final refined mesh of the respective method for 5 different refinement levels on 3 randomly selected PDEs. For the RL methods, all policies are taken from the first repetition of the 10 random seeds conducted for the respective experiment.

## G.1   ASMR Refinements

We visualize exemplary refinements of ASMR policies for all considered tasks in Figures 18 (Laplace's equation), 19 (Poisson's equation), 20 (Stokes equation), 21 (Linear Elasticity), and 22 (Heat Diffusion). Across all tasks, ASMR is able to provide highly accurate refinements for different numbers of total elements.

## G.2 Baseline Comparisons

Figure 23 shows refinements for *Single Agent* for different total timesteps $T$, Figure 24 presents *VDGN* with different $\alpha$ values. Figure 25 visualizes refinements of *Sweep* for a varying number of maximum elements $N_{\max}$. Figures 26, 27 and 28 show refinements of the *Oracle*, *Maximum Oracle* and *ZZ Error Heuristics* for different values of the threshold $\theta$.

The visualizations show that the RL baselines struggle to provide consistent high-quality refinements for different mesh resolutions. The *Single Agent* baseline sometimes focuses on uninteresting regions of the mesh or refines the same area too often. The *VDGN*-like baseline performs well in some cases, but collapses to no refinements or fully uniform refinements for some evaluation PDE. *Sweep* provides almost uniform refinements for most PDEs and element budgets, likely as a result of the misalignment in the environment transitions between training and inference. The *Heuristics* greedily refine the elements with the largest error estimates in their respective metric, regardless of the resulting decrease in error. This behavior generally leads to locally accurate refinements, but fails to effectively decrease the mesh error in some cases. Additionally, the heuristics act locally, which causes potential issues for PDEs with global dependencies [92] and conforming refinements. The *ZZ Error Heuristic* sometimes misses regions of interest, but provides a much smoother refinement than the *Oracle Error Heuristic*, which can be beneficial for the error reduction in some cases.

## G.3 Element Markings

Figure 29 visualizes a full rollout of our method, including the markings of the elements after every step.

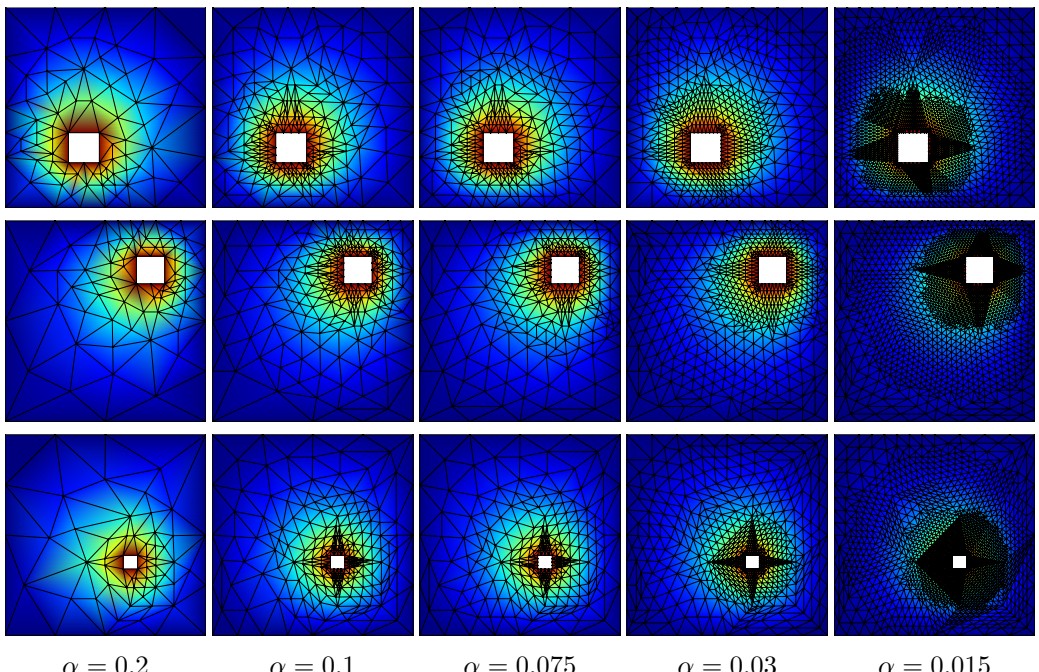

$\alpha = 0.2$     $\alpha = 0.1$     $\alpha = 0.075$     $\alpha = 0.03$     $\alpha = 0.015$

Figure 18: Final refined meshes of ASMR for randomly sampled PDEs for the Laplace equation for different element penalties $\alpha$.

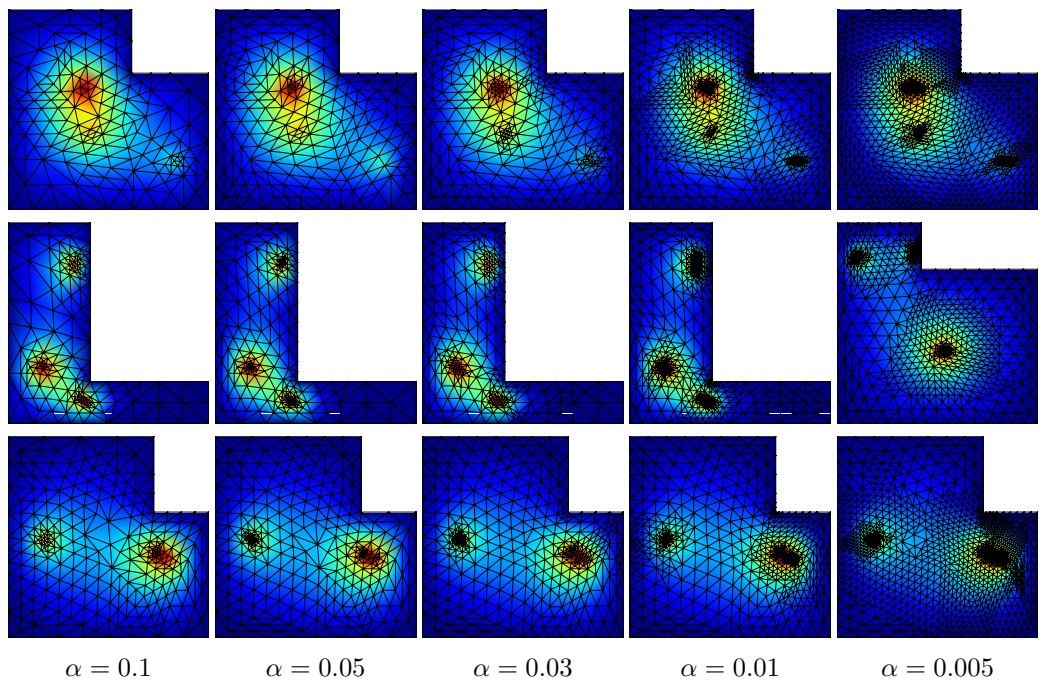

Figure 19: Final refined meshes of ASMR for randomly sampled PDEs for the Poisson equation for different element penalties $\alpha$.

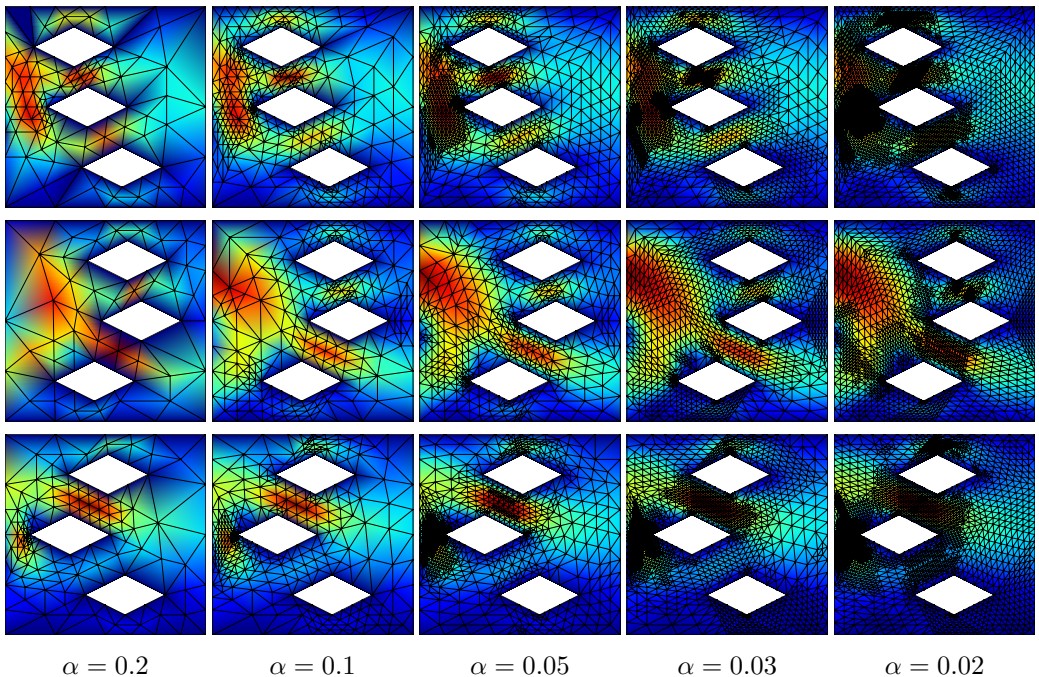

Figure 20: Final refined meshes of ASMR for randomly sampled PDEs for the Stokes flow task for different element penalties $\alpha$.

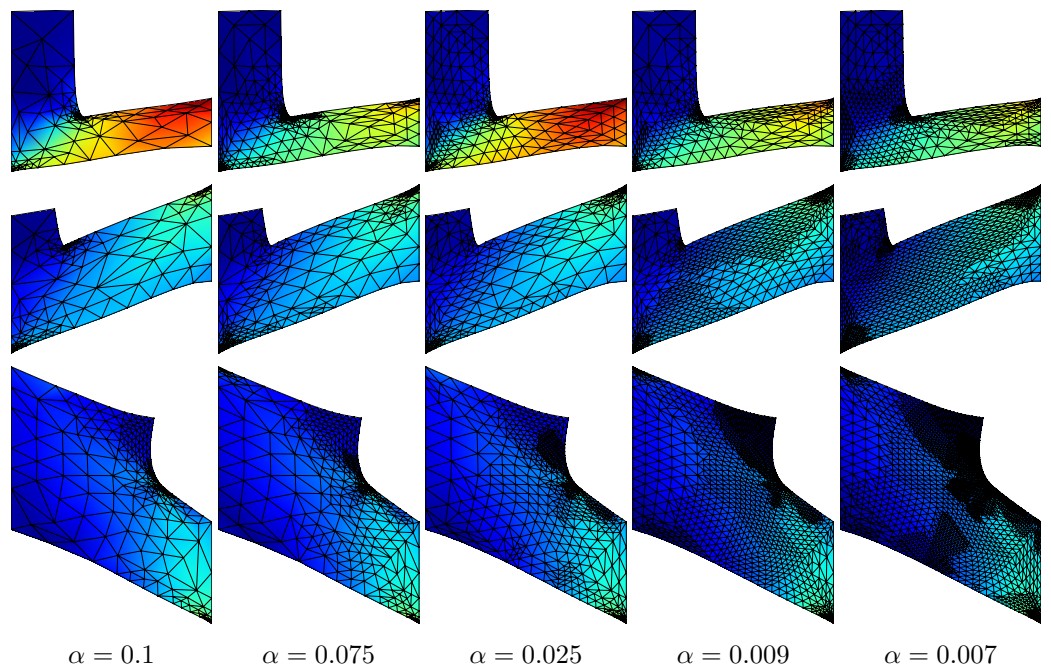

$\alpha = 0.1$     $\alpha = 0.075$     $\alpha = 0.025$     $\alpha = 0.009$     $\alpha = 0.007$

Figure 21: Final refined meshes of ASMR for randomly sampled PDEs for the linear elasticity task for different element penalties $\alpha$. The visualizations show the deformed meshes, which are originally L-shaped.

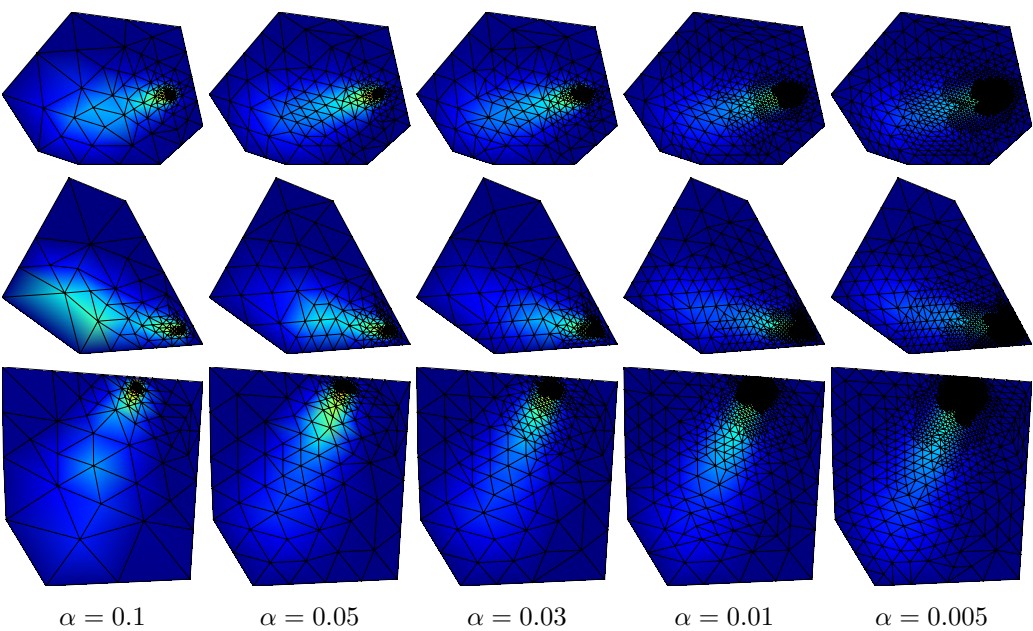

$\alpha = 0.1$     $\alpha = 0.05$     $\alpha = 0.03$     $\alpha = 0.01$     $\alpha = 0.005$

Figure 22: Final refined meshes of ASMR for randomly sampled PDEs for the non-stationary heat diffusion task for different element penalties $\alpha$.

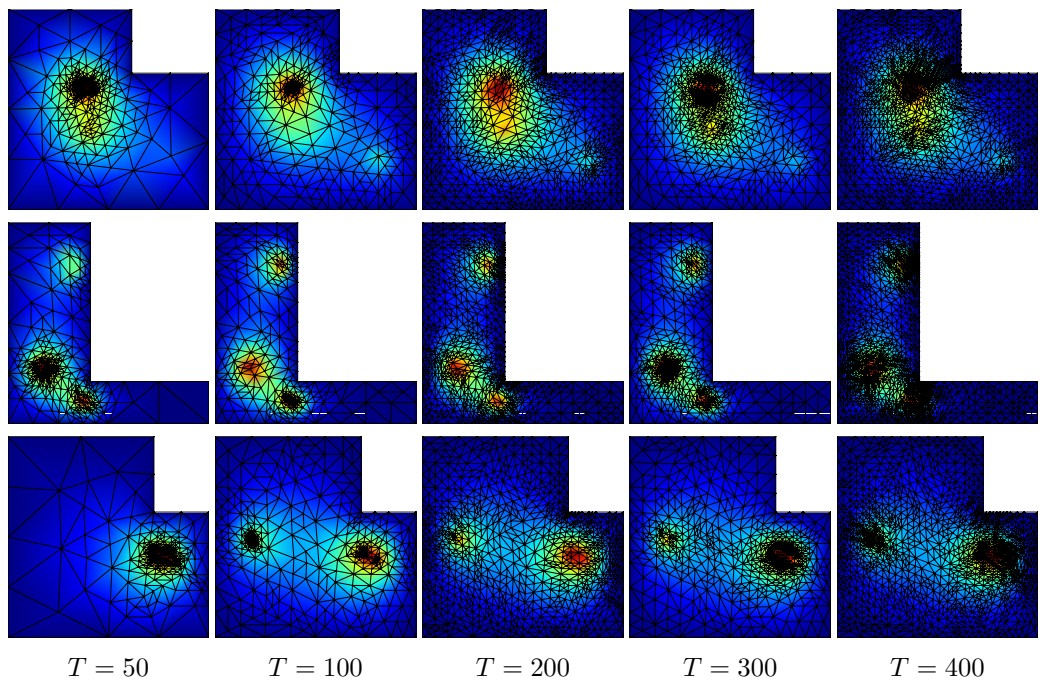

Figure 23: Final refined meshes of the *Single Agent* baseline for the Poisson equation on randomly sampled PDEs for different environment rollout lengths $T$.

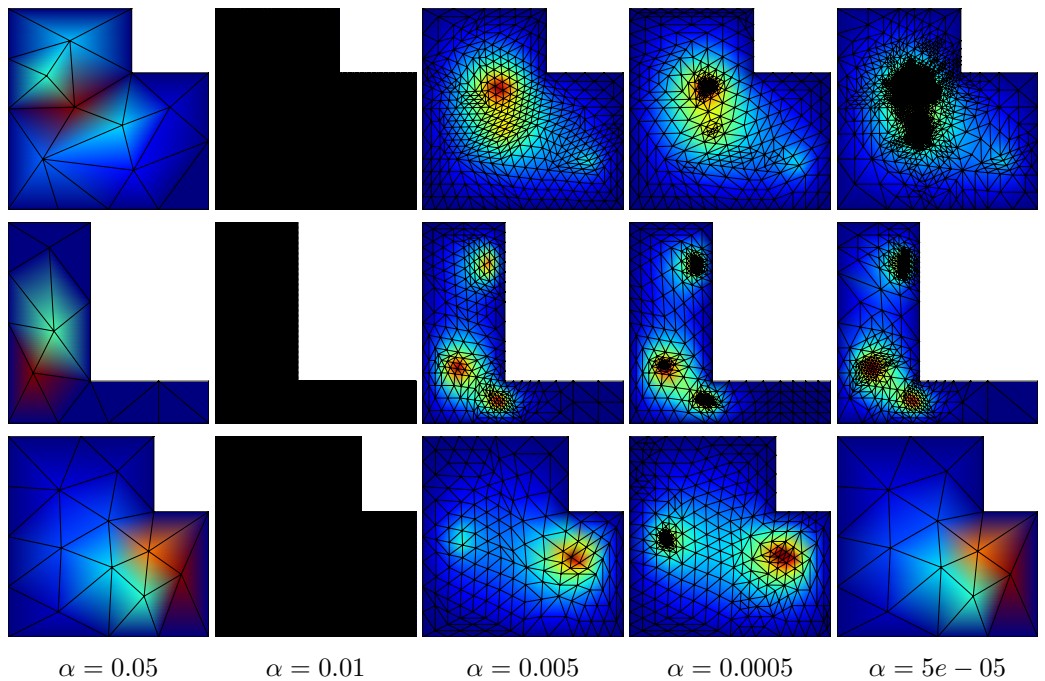

Figure 24: Final refined meshes of the *VDGN*-like baseline for the Poisson equation on randomly sampled PDEs for different element penalties $\alpha$.

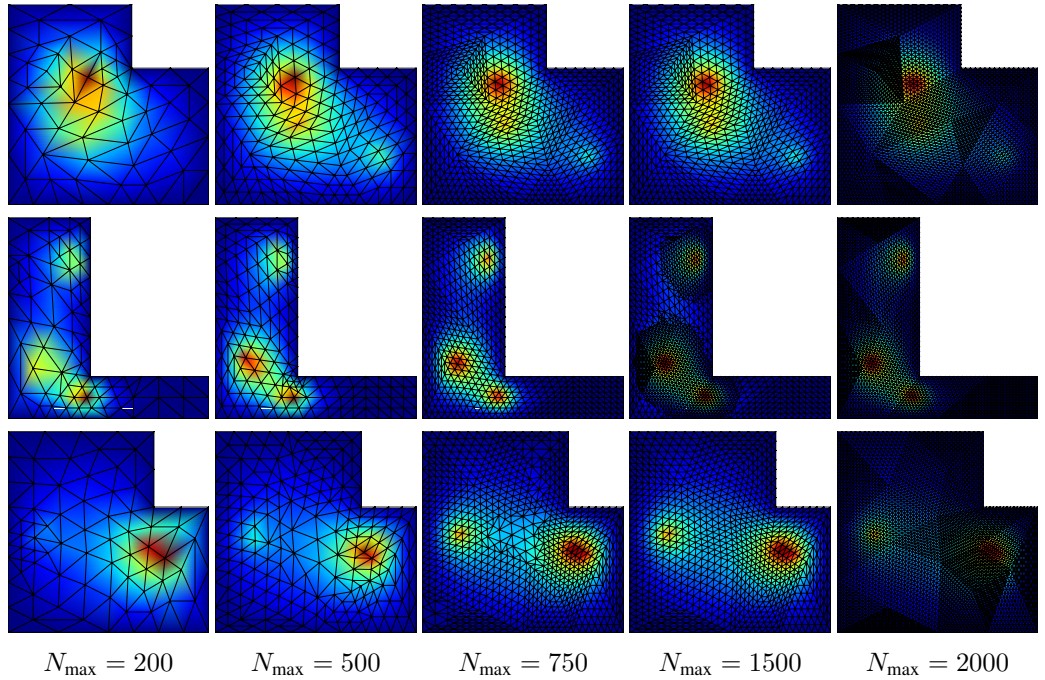

| $N_{\max} = 200$ | $N_{\max} = 500$ | $N_{\max} = 750$ | $N_{\max} = 1500$ | $N_{\max} = 2000$ |

Figure 25: Final refined meshes of the *Sweep* baseline for the Poisson equation on randomly sampled PDEs for different maximum numbers of elements $N_{\max}$.

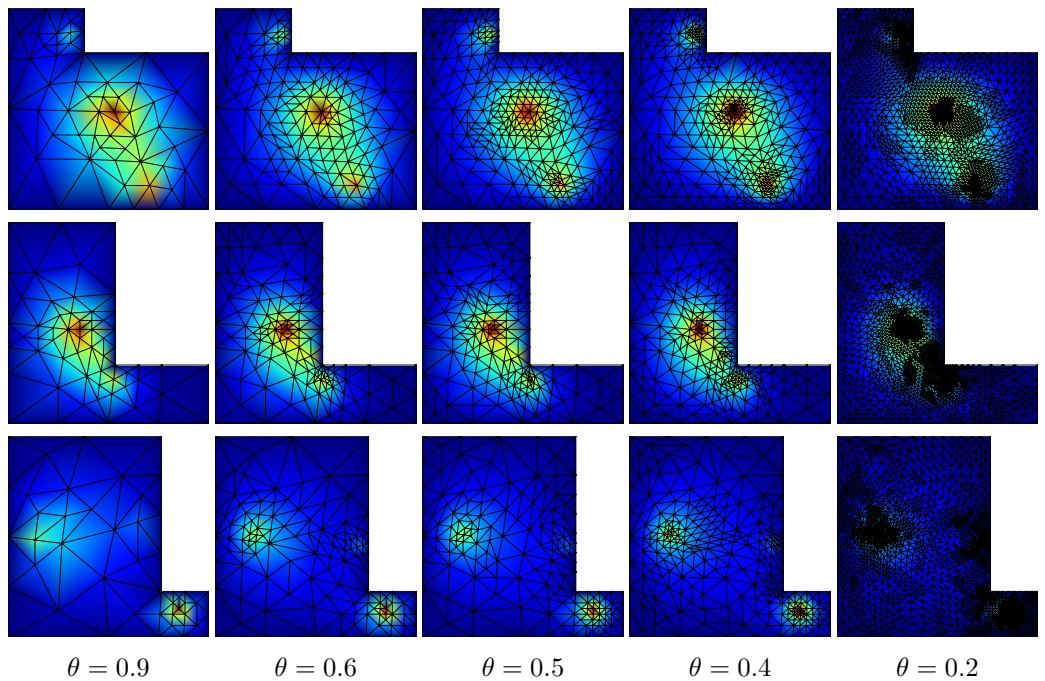

| $\theta = 0.9$ | $\theta = 0.6$ | $\theta = 0.5$ | $\theta = 0.4$ | $\theta = 0.2$ |

Figure 26: Final refined meshes of the *Local Oracle* baseline for the Poisson equation on randomly sampled PDEs for different error thresholds $\theta$.

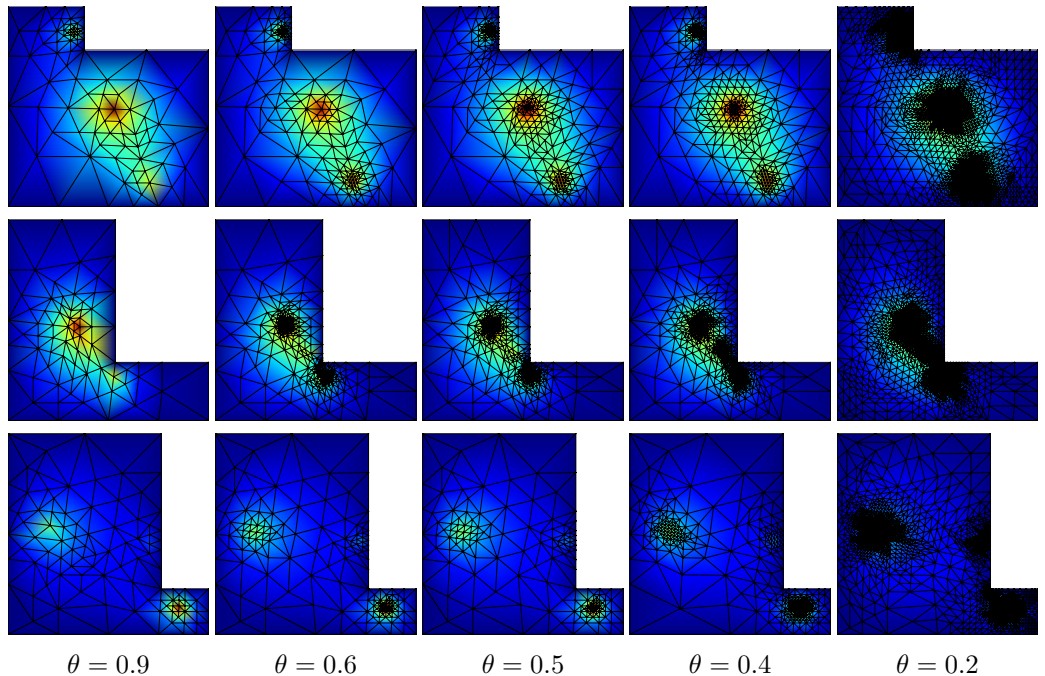

Figure 27: Final refined meshes of the *Local Maximum Oracle* baseline for the Poisson equation on randomly sampled PDEs for different error thresholds $\theta$.

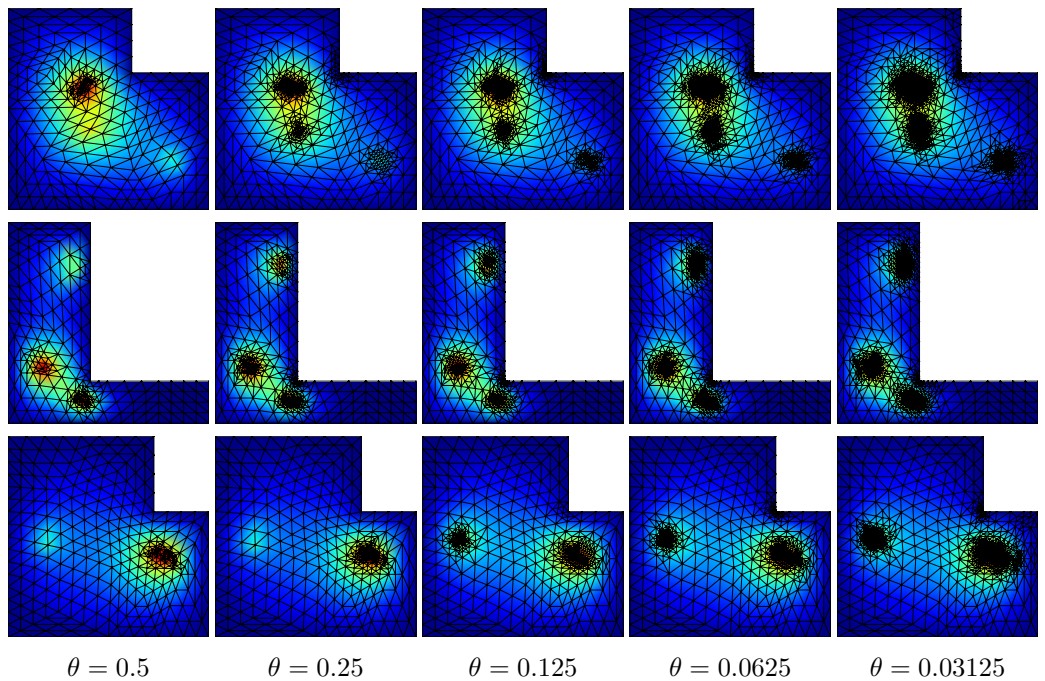

Figure 28: Final refined meshes of the *ZZ Error* for the Poisson equation on randomly sampled PDEs for different error thresholds $\theta$.

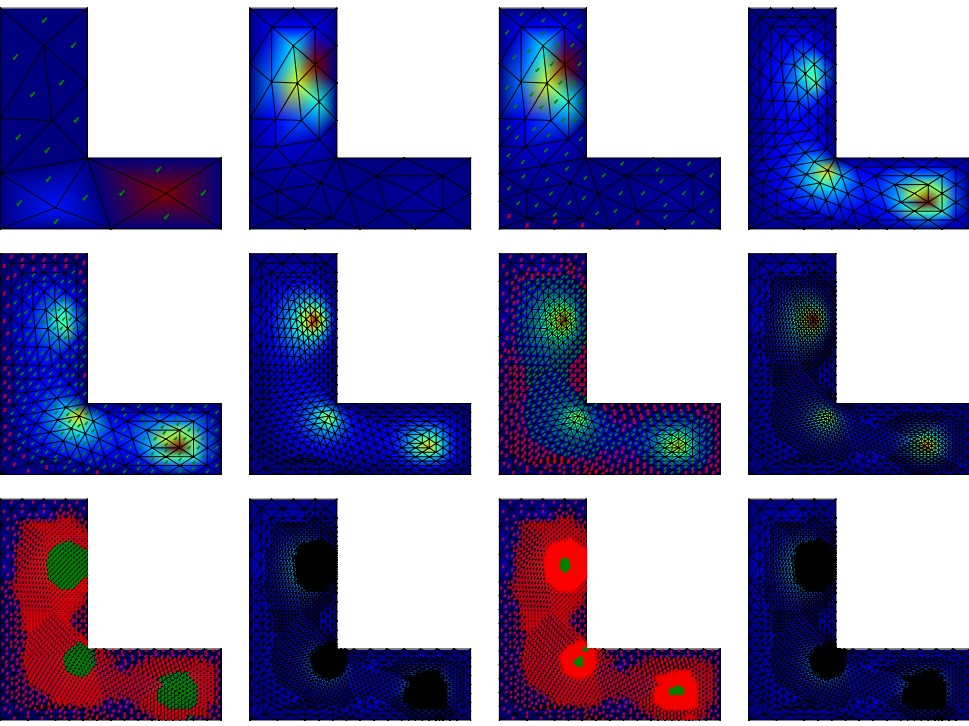

Figure 29: Visualization of a full rollout of our method on a Poisson task, including the markings of the elements after every step. The figures in the first and third row show the markings, and the second and fourth row show the resulting refined meshes.

