# OpenReview forum: "Swarm Reinforcement Learning for Adaptive Mesh Refinement"
_NeurIPS.cc/2023/Conference — NeurIPS 2023 poster_

### Official Review · Reviewer_qjoP · 2023-06-11

**Soundness:** 2 fair
**Presentation:** 3 good
**Contribution:** 2 fair
**Rating:** 4
**Confidence:** 5

**Summary:**

This paper formulates h-adaptive mesh refinement (AMR) as a decentralized partially-observable Markov decision process (Dec-POMDP), and proposes a methods ASMR with parameter-sharing among agents and individual rewards to find refinement strategies. Refinement policies are parameterized by message-passing graph neural networks (MPN) and trained by Q-learning (specifically, DQN) and proximal policy optimization (PPO). Evaluation was conducted on various 2D elliptic partial differential equations (PDEs) with different domains, showing that the proposed approach is competitive with traditional error-based AMR heuristics and outperforms modified versions of existing learning-based approaches.

**Strengths:**

Originality: One main novelty of this paper is the use of independent multi-agent reinforcement learning with individual rewards/returns for AMR. The proposed individual return is a novel solution to the posthumous credit assignment problem that arises in the AMR context, whereby an agent vanishes upon taking a refinement action. It also has the advantage of avoiding the difficulties of the regular multi-agent credit assignment problem that arises from using a single team reward.

Quality: Overall, this paper covers most of the aspects of a high-quality applied paper, as it provides sufficient motivation and background for the application and provides a rigorous formulation and empirical evaluation.

Clarity: The description of the problem, the proposed method, the notation, and experimental setup and results are all clearly written. See below for more comments about clarity.

Significance: The results in this paper are significant as they provide a path to improving AMR for the finite element method, which is a critical tool in engineering and applied sciences for solving PDEs.

**Weaknesses:**

The characterization of the scale of experiments in this paper, in contrast to previous work, should be made more precise. The abstract of this paper says that previous work "scale only to simple toy examples", which leads readers to expect that this paper deals with real-world problems on large domains. However, the experiments in this paper were conducted only on simple elliptic PDEs on meshes with up to thousands of elements, which appears to be on the same level of complexity and scale as previous work (e.g., AMR on simple hyperbolic equations with up to thousands of elements). The paper's Limitation section also acknowledges this fact, so the abstract and main paper should be made consistent.

The proposed approach seems to be extendable to include coarsening actions, but this is not demonstrated in this paper and hence it is hard to judge whether the proposed approach holds promise if extended. There is existing work that support coarsening (refs 28,29), so one would expect this to be a requirement for this application area. One would expect that an optimal strategy for the moving heat source problem requires coarsening actions, since regions that were refined due to proximity to the source may need to be coarsened once the source has moved away from them.

Experimental evaluation includes randomization over domains, boundary conditions and initial conditions. However, for trained policies to be useful in applications to real-world problems, one should show that trained policies can work on larger meshes and longer step counts than those seen in training. The proposed method uses global features such as the number of mesh elements and environment time step, which this leads to concerns about what would happen if the trained policies are run on test cases where the number of elements and step count go out of the training distribution.

It was confusing to read the phrase "PPO version of VDGN" (line 245) and "VDGN...produce better results with PPO" (line 265), since PPO is a policy optimization method that is fundamentally different from value-based methods, so it is unclear how exactly PPO can be used in combination with VDGN. Upon looking closer, it appears that some of the learning-based baselines described in Section 4 do not match the methods in the previous work (refs 27,28), so it is misleading to label them as such. For the "Argmax" method, (ref 27) trains a stochastic policy on a discrete action space consisting of all the available elements, but the implementation in this paper "predicts a continuous action for each mesh element." It is not clear what is that continuous action and what RL algorithm is used to train this baseline. For the baseline labeled as "VDGN", it appears that the implementation in this paper is an actually an on-policy policy optimization method based on PPO, using a decomposed value function. But that is fundamentally different from VDGN in (ref 28), which looks like an off-policy method based on Q-learning.

Additional suggestions:
- line 37: "shared observations" imply that agents have the same observation, but that is not the case since each agent is a node in the graph and have different observations. Perhaps the authors mean to say agents have the same observation space.
- It is hard to see what is gained by calling the formulation an "Adaptive Swarm Markov Decision Process", as opposed to the more standard terminology of a Dec-POMDP with parameter-sharing among agents.
- line 257: not clear what is meant by "holistic" in this context. Perhaps the authors mean that RL can find optimal strategies for long-term objectives

**Questions:**

Suggestions:
- Demonstrate that trained policies generalize to larger meshes and more time steps than those in training. This is necessary for policies trained on tractable small problems to be deployed in real FEM simulations.
- some of the learning-based baselines implemented in this paper do not accurately reflect the methods in the referenced papers, so they should not be described as though they represent those works.
- Include empirical evidence that the approach extends to coarsening actions.

**Limitations:**

The statement of limitations should include the absence of empirical results on coarsening, and the lack of other classes of PDEs such as hyperbolic equations.

---

> ### Author Rebuttal · Authors · 2023-08-09
>
> We thank the reviewer for their detailed feedback, particularly noting the innovative use of swarm reinforcement learning for AMR and the comprehensive clarity and rigor with which we presented our findings. We now address individual concerns, including the clarification needed in the experimental section and the accurate representation of related work.
>
> > […] the experiments in this paper were conducted only on simple elliptic PDEs on meshes with up to thousands of elements, which appears to be on the same level of complexity and scale as previous work. […] the abstract and main paper should be made consistent
> >
>
> Our experiments use conforming meshes and more refinement levels than existing work. While the considered PDEs are relatively simple, the difficulty of finding good refinements for them is evidenced by the performance of the baseline methods. We provide further discussion in the general answer and will address this distinction in the revision. We also refer to Figure 2 of the accompanying PDF for a refinement with more than $50\,000$ elements.
>
> > One would expect that an optimal strategy for the moving heat source problem requires coarsening actions […] Include empirical evidence that the approach extends to coarsening actions.
> >
>
> We solve the Heat Diffusion equation in an inner loop and optimize the mesh with respect to the error of the last step of the equation as described in Appendix B.5. Respecting the movement of the heat source is still important due to the propagation of errors, and this task showcases that our method is able to create a single refined mesh that provides accurate results for the full simulation.
>
> We agree that mesh coarsening is important for general AMR strategies and aim to explore this direction further in future work. We believe that iterative refinement from a coarse initial mesh is sufficient for the problems considered in this work, which we discuss in more detail in the general answer.
>
> > […] one should show that trained policies can work on larger meshes and longer step counts than those seen in training. The proposed method uses global features […] Demonstrate that trained policies generalize to larger meshes and more time steps than those in training.
> >
>
> Figure 2 of the general PDF shows that ASMR, with minor modifications to the training environment and observation space (excluding, in particular, the global information), can generalize to meshes with over $50\,000$ elements on the Poisson problem, achieving a speedup of more than $100$ times when compared to $\Omega^*$. We equate the initial size of elements in this larger mesh to those seen during training so that the six refinement steps that are used during training are sufficient.
>
> > It was confusing to read the phrase "PPO version of VDGN" (line 245) […] For the "Argmax" method, (ref 27) trains a stochastic policy on a discrete action space consisting of all the available elements […] Some of the learning-based baselines implemented in this paper do not accurately reflect the methods in the referenced papers […]
> >
>
> In Figure 8 in Appendix D.1, we compare VDGN (Ref. 28) with DQN and our “VDQN+PPO” variant. The latter is a policy gradient method and can not directly apply the value decomposition in the Q function. However, since PPO uses a value function as a baseline to approximate the advantage function, we apply a similar value decomposition there (c.f. Line 245). We find in Figure 8 that VDGN (DQN) performs significantly worse than our PPO variant, which is why we use the latter for the main experiments in the paper.
>
> *Argmax* (Ref. 27) trains a discrete policy that chooses one of the mesh elements for refinement. This is realized via a categorical distribution over all elements, which is computed via a softmax over the policy outputs for each element. We experiment with both DQN and PPO for our *Argmax* baseline in Figure 8 in Appendix D.1. For the PPO version, we take the maximum over a continuous scalar per element (c.f. Line 239). Since the softmax is order-preserving, this corresponds to acting on a categorical distribution. We directly use a categorical distribution for the DQN variant and apologize that the paper does not state this clearly.
>
> In summary, “VDGN+PPO” uses the value decomposition idea of VDGN for its advantage estimation, while the Argmax baseline essentially uses a categorical distribution to select the next element to be refined. We thank the reviewer for pointing out the issues with the phrasing and nomenclature for the baselines and will adapt both in the revision.
>
> > line 37: "shared observations" imply that agents have the same observation […]. Perhaps the authors mean to say agents have the same observation space.
> >
>
> We agree that “shared observation space” in Line 37 is technically more correct. We originally chose to use “shared observations” here due to the overlapping receptive fields of the individual agents and will adapt this in the revision.
>
> > It is hard to see what is gained by calling the formulation an "Adaptive Swarm Markov Decision Process" […]
> >
>
> We avoid the DEC-POMDP formulation due to agent-wise rewards, changing action and observation spaces and crucially mappings between the agents over time, all of which can be more naturally represented when formulating AMR as a swarm problem. We refer to the general answer for further details.
>
> > line 257: not clear what is meant by "holistic" in this context. Perhaps the authors mean that RL can find optimal strategies for long-term objectives
> >
>
> “Hollistic” refers to both the optimization of long-term objectives, and to finding an optimal refinement w.r.t. the full receptive field of the agent rather than just the element itself.
>
> We hope that this answer addresses the concerns of the reviewer. If any reservations or questions about the paper remain, we encourage the reviewer to reach out to us for further clarification.

---

> > ### Comment · Reviewer_qjoP · 2023-08-17
> > **Acknowledgement of rebuttal, main concerns on complexity claim and lack of coarsening remain; question remains about whether baselines match prior work.**
> >
> > Regarding claim of complexity: I appreciate that the author ran an additional experiment on more than 50k elements shown in Figure 2 of the rebuttal PDF. However, this single result by itself is not sufficient to justify the way the abstract was written, which makes it sound as though the overall complexity of experiments in this paper is another level higher than previous work. The mostly stationary problems and the smooth nature of heat diffusion considered in this paper are arguably simpler than time-dependent problems where the solution feature cross over multiple elements between re-mesh steps.
> >
> > Regarding lack of coarsening and the heat diffusion problem: The lack of coarsening is a significant weakness, since it is critical for more complex time-dependent PDEs with rapidly moving features beyond the pedagogical problems used in this paper. The smoothness of the heat diffusion problem used in this paper is not a good representative example of the level of difficulty of time-dependency that needs to be handled in other problems, e.g. Euler equations.
> >
> > Regarding generalization to larger meshes: I appreciate the new result, and I believe this needs to included in the main paper because this kind of generalization is a necessary condition for using RL-trained AMR policies in practice.
> >
> > Regarding whether or not the baselines in this paper match exactly with prior work: I appreciate the clarification by the authors. For the baseline labeled "Argmax", is the use of PPO in this paper the same as the use of PPO in the ref [27]? In ref [27], the method has a global action space over all elements, and PPO is used as the single-agent RL for this action space. Is this the case for the "PPO implementation of Argmax" in this paper? Or is the PPO applied independently for each individual element (akin to the decentralized PPO used for the proposed method in this paper)? If applied independently, then this quite different from ref [27]. For the baseline labeled "VDGN", if the implementation shown in the main paper is the "PPO version" created by the authors, then it is necessary to revise the label names in Figure 4 and 5 to make the distinction clear. Upon closer reading, I see the authors use an MPN for the baseline labeled "VDGN", whereas ref [28] has a graph attention network. This architectural difference alone makes it hard to label the baseline as "VDGN".

---

> > > ### Author Response · Authors · 2023-08-18
> > >
> > > We sincerely thank the reviewer for the additional discussion.
> > >
> > > > I appreciate that the author ran an additional experiment on more than 50k elements shown in Figure 2 of the rebuttal PDF. However, this single result by itself is not sufficient to justify the way the abstract was written, which makes it sound as though the overall complexity of experiments in this paper is another level higher than previous work. [...]
> > >
> > > We thank the reviewer for appreciating the additional experiments. Regarding the experiments' complexity, we concentrate on mostly stationary problems as these problems are important and ubiquitous in engineering [1,2,3,4]. Here, the difficulty lies in finding multiple accurate refinement steps. Our experiments show that all methods can do so for small meshes and shallow refinements (c.f., Figures 4, 5), but that only ASMR consistently provides good refinements for larger meshes. In the revised paper we will attenuate our claim of increased complexity to clarify the focus on finding multiple accurate refinement steps. The increase in complexity of our experiments with respect to these types of refinement compared to previous work is evident as the performance of the baselines breaks down with an increasing number of mesh elements.
> > >
> > > > The lack of coarsening is a significant weakness, since it is critical for more complex time-dependent PDEs [...]
> > >
> > > Our work focuses on conforming AMR due to its importance and prevalent use in engineering. While coarsening for conforming meshes is very difficult due to the non-local nature of conforming refinements (c.f. Ref. 41), they offer more accurate solutions and increased stability of the underlying system, making them the prevalent choice in many applications with stationary solutions [1,2,3]. Our experiments clearly show that our algorithm outperforms existing RL-based algorithms for this crucial use-case. We agree that mesh coarsening is an interesting aspect of AMR with non-conforming meshes, and briefly discuss how ASMR can be extended accordingly in Line 165.
> > >
> > > > Regarding generalization to larger meshes: I appreciate the new result, and I believe this needs to included in the main paper because this kind of generalization is a necessary condition for using RL-trained AMR policies in practice.
> > >
> > > We appreciate that the reviewer agrees that the new generalization results are significant. We will gladly include these results and an additional discussion on generalization in the revised main paper.
> > >
> > > > For the baseline labeled "Argmax", is the use of PPO in this paper the same as the use of PPO in the ref [27]? [...]
> > >
> > > We thank the reviewer for bringing this to our attention. We implemented the Argmax method as described in Ref 27 and found in preliminary experiments that the results are similar to those reported in our paper. In general, the method yields acceptable results on small instances, but does not scale to larger meshes and numbers of refinements. Note that Ref 27 optimized the hyperparameters on a per-task basis, which makes it more difficult to use in practical scenarios. We instead tuned one set of hyperparameters per method across all tasks.
> > >
> > > > For the baseline labeled "VDGN" [...] it is necessary to revise the label names in Figure 4 and 5 to make the distinction clear. [...] the authors use an MPN for the baseline labeled "VDGN", whereas ref [28] has a graph attention network. This architectural difference alone makes it hard to label the baseline as "VDGN".
> > >
> > > We consistently employ MPNs for all RL-learned baselines to ensure a fair comparison between methods. The contribution of VDGN lies in framing AMR as a multi-agent problem and applying a value decomposition to the RL objective. These advancements are independent of the underlying GNN architecture. We show in our experiments that while the value decomposition works well for small instances, it struggles to provide a good reward signal for larger meshes. We expect Graph Attention Networks to show a similar performance to MPNs, and will include them as an ablation in the revision. We will also revise the labels and clarify that the baseline implements the important aspects of VDGN, but differs in some details for the sake of a fair comparison.
> > >
> > > We want to thank the reviewer again for the continued discussion and will gladly provide further clarifications if requested.
> > >
> > > [1] Nagarajan, A., & Soghrati, S. (2018). Conforming to interface structured adaptive mesh refinement: 3D algorithm and implementation. *Computational Mechanics*
> > >
> > > [2] Ho-Le, K. (1988). Finite element mesh generation methods: a review and classification. *Computer-aided design*
> > >
> > > [3] Jones, M. T., & Plassmann, P. E. (1997). Adaptive refinement of unstructured finite-element meshes. *Finite Elements in Analysis and Design*
> > >
> > > [4] Geuzaine, C., & Remacle, J. F. (2009). Gmsh: A 3‐D finite element mesh generator with built‐in pre‐and post‐processing facilities. *International journal for numerical methods in engineering*

---

### Official Review · Reviewer_LCc4 · 2023-07-05

**Soundness:** 3 good
**Presentation:** 4 excellent
**Contribution:** 3 good
**Rating:** 6
**Confidence:** 3

**Summary:**

This paper proposes a novel MDP formulation and policy architecture for adaptive mesh refinement.
The MDP formulation (ASMDP) defines the components of the MDP in order to account for the changing number of agents across timesteps, and the reward is formulated in a manner to make credit assignment easier and to account for particular aspects of the AMR task (e.g., accounting for the area associated with an element).
The paper applies RL algorithms (PPO, DQN) with a graph neural network policy (with a discrete action space of whether or not to refine a particular node).
The experiments consider a set of 2d mesh refinement problems and evaluate primarily with top 0.1% error against other learned and oracle baselines, showing that ASMR outperforms the other learned baselines.


**Strengths:**

### Significance
- Problem setting: FEM plays an important role in many fields and (given the potential scale of such problems) improvements in the accuracy / computational efficiency could have a significant impact on the rate of progress or computation required
- Contribution: The proposed problem formulation / algorithm significantly improves performance and reliability on a sample of AMR tasks as demonstrated in the experiments and appendix, which constitutes a significant contribution.
### Originality
- Aspects of the MDP formulation (reward function, aspects of the handling of a time-varying number of agents) appear to be novel contributions.
- The use of the particular policy architecture appears to be novel (though the use of GNNs for the policy isn’t novel)
- Extensive experiments investigating the effectiveness of the proposed formulation/algorithm in a variety of of AMR setting (demonstrating sota performance of the algorithm)
### Quality
Approach
- The application of RL in this setting is reasonable. The problem formulation / algorithm are formulated reasonably.
Evaluation
- The main experimental claim is that ASMR outperforms existing learned methods for AMR, and this is evaluated extensively through experiments across different tasks with appropriate metrics, and the results support the claim.
- A number of important additional questions are addressed experimentally by the paper (e.g., ablations, use of top 0.1% error directly as reward, runtime comparison w/ just computing $\Omega^{*}$)
### Clarity
- The paper is exceptionally well-written and clear. Figures and tables are explained clearly and contribute to the understanding of the method / experiments.
- Sufficient details are given in the appendix to reproduce the paper, and code is provided in the supplement, which appears to be reasonably well-designed / implemented.


**Weaknesses:**

### Originality
Algorithm
- Ultimately many aspects of the problem formulation / algorithm are based on prior work, and the novel contributions are (1) specific aspects of the problem formulation (e.g., reward function), (2) the application of existing methods to the problem (some of which are novel) (e.g., policy architecture, RL algorithms), and (3) extensive experimental evaluation.
### Quality
Algorithm
- See questions section
Evaluation
- Further evaluation of the generalization abilities of the algorithm would be beneficial (e.g., building on D.4 with quantitative metrics) because (I assume) that an RL algorithm for AMR would in practice be applied outside its training distribution.
- Relatedly, it’s not clear to me based on the experimental results that practitioners using AMR for FEM would prefer ASMR to the local oracle or local maximum oracle approaches in practice (perhaps ASMR is faster excluding training, but is the reduction in simulation time worth the less consistent performance?). Does increasing the amount of training data (or other parameters) improve the performance of ASMR beyond these oracle baselines across (more of) the tasks?
### Clarity
- The setting / setup of section D.4 on the generalization capabilities is unclear to me


**Questions:**

Problem formulation / algorithm
1. Does it make sense to formulate the problem as a SwarMDP / special case of a DecPOMDP? It seems that in practice (both at training and inference time) the problem is fully observable and decision making is performed in a centralized fashion (which would make it an MDP). Is that not the case? Perhaps this is sort of a semantics question about what constitutes the problem formulation vs solution.
2. Relatedly, assuming the SwarMDP formulation, in what sense is the problem partially observable to individual agents/elements? It seems that each agent makes decisions based on an observation space capturing information about the full state of the system (i.e., the observation graph taken as input seems to fully capture the state of the system). Is that not the case?
3. Given that $\Omega^{\*}$ and $\Omega^{0}$ are available, it seems like the problem could be formulated as a supervised learning problem (SL for AMR is discussed in the related work, but I don’t think the references answer the following question). Given this, why is it necessary to formulate the problem as an RL problem? Is it simply that mapping from $\Omega^{0}$ to $\Omega^{*}$ is best done in an iterative manner (due to the variable number of nodes) (thereby making it a sequential decision making problems best addressed by RL)?

Evaluation
1. In the runtime comparison of section D.3, it seems like either of the processes being compared could be parallelized effectively on the GPU. Is that not the case? Why is the comparison performed on CPUs?
2. Related to that runtime comparison, what is the additional runtime cost of executing the local oracle approaches? Does it make sense to consider that in the comparison?
3. In figure 14 (right), would you elaborate on why ASMR underperforms the baselines?


**Limitations:**

See weaknesses and questions sections

---

> ### Author Rebuttal · Authors · 2023-08-09
>
> We are grateful to the reviewer for their detailed evaluation of our paper, particularly emphasizing our method's significant contributions and robust experimental claims. We will now respond to the individual concerns raised by the reviewer, including the generalization capabilities of the method and the importance of framing AMR as a Swarm Reinforcement Learning problem.
>
> > Further evaluation of the generalization abilities of the algorithm would be beneficial (e.g., building on D.4 with quantitative metrics) […].
> >
>
> We recognize the importance of robust generalization for learned AMR methods and show further qualitative experiments in Figure 2 of the general PDF. Initial quantitative evaluations suggest that generalization is consistent across methods for all domains shown in Figure 11, likely due to the local perspective that all methods use. Given ASMR's superior performance, it also seems to perform best in these generalization tests. We will detail these quantitative evaluations in the revision.
>
> > perhaps ASMR is faster excluding training, but is the reduction in simulation time worth the less consistent performance?
> >
>
> Figure 10 in the Appendix shows that our method is up to $30$ times quicker than uniform refinement, while the main experiments underscore its improved reliability compared to other RL-driven approaches. To iterate on this, the large mesh in Figure 2 of the general PDF takes $10$-$12$ seconds to produce, whereas the uniform refinement needs $~20$ *minutes*. Training ML models such as ASMR in early stages of engineering development thus allows to speed up subsequent phases due to faster inference times. Generalizing well also allows for future problems to be predicted, making the training time negligible.
>
> > Does increasing the amount of training data (or other parameters) improve the performance of ASMR beyond these oracle baselines across (more of) the tasks?
> >
>
> Figure 9 in Appendix D.2 illustrates that more training PDEs marginally boost performance. Our preliminary results indicate that larger latent dimensions offer slight performance gains. Since our main goal is to introduce and assess an effective RL framework for AMR, we do not fine-tune the specific architecture to surpass the oracle heuristics. Still, we acknowledge the importance of this optimization for practical usage and will discuss it in more detail in our paper's revised version.
>
> > The setting / setup of section D.4 on the generalization capabilities is unclear to me
> >
>
> In Appendix D.4, we apply a singular trained policy to problems outside the training distribution. The policy's training data comprises 100 L Shapes with three-component load functions, as shown in the middle-left part of Figure 11. The rest of Figure 11 depicts ASMR's generalization across various shapes (left to right) and load functions (top to bottom). We'll offer more clarity in our paper's updated version.
>
> > Does it make sense to formulate the problem as a SwarMDP / special case of a DecPOMDP?  It seems that […] the problem is fully observable […] in what sense is the problem partially observable to individual agents/elements?
> >
>
> We discuss the importance of the adaptive swarm setting in the general answer. While the problem is fully observable, each agent's observation space is localized, with the receptive field depending on the number of message passing steps of the GNN. Each agent can access a shared global vector of limited size (32 in our experiments), keeping the observation space predominantly local.
>
> > Given that Ω∗ and Ω0  are available, it seems like the problem could be formulated as a supervised learning problem […] why is it necessary to formulate the problem as an RL problem?
> >
>
> We present AMR as an RL problem because of its sequential nature, noting that it additionally allows for non differentiable refinement and PDE-solving processes, which increase the methods applicability. This approach aligns with existing literature (Refs. 27-29), whereas supervised methods often target error estimators (Refs.71-72) or specific AMR facets (Refs.73-74) instead of actual refining strategies directly. We'll clarify this distinction in our revised related work section.
>
> > […] it seems like either of the processes being compared could be parallelized effectively on the GPU.
> >
>
> Since our experiments utilize Scikit-FEM (Ref. 39) which is CPU-only, we perform all runtime tests on a CPU for fairness. In general, both the policy and solving the PDE could be efficiently parallelized on a GPU. Broadly speaking, our model's scalability is approximately linear with the refined mesh's elements, while solving a uniform refinement becomes progressively costlier as the problem size increases.
>
> > What is the additional runtime cost of executing the local oracle approaches?
> >
>
> The cost of executing the local oracle heuristics is dominated by the calculation of the reference refinement $\Omega^*$. Since only computing $\Omega^*$ already takes longer than our policy does, we chose not to include them in Figure 10. We will clarify this in the revision.
>
> > In figure 14 (right), would you elaborate on why ASMR underperforms the baselines?
> >
>
> In Figure 14, ASMR shows good performance for large meshes but lags behind some baselines for smaller instances. This is mainly because ASMR skips refining larger elements far from the flow's inlet. While these elements don't show a significant maximum error (as seen in Figure 5's left side), they contribute to a considerable total error mass. Many baselines offer relatively uniform refinements, explaining their performance and similarity with the uniform comparison. Crucially, ASMR recognizes that errors on the left side intensify and accumulate as information flows to the right, explaining its effectiveness for larger element counts.
>
> We hope that our answer clarifies some of the concerns of the reviewer, and would be grateful for further communication if any issues persist or new ones arise.

---

### Official Review · Reviewer_ymhS · 2023-07-06

**Soundness:** 3 good
**Presentation:** 2 fair
**Contribution:** 3 good
**Rating:** 6
**Confidence:** 3

**Summary:**

This paper builds on recent advances in learned adaptive mesh refinement method to scale to complex physical simulations. Instead of formulating AMR as a reinforcement learning problem with a single agent, the authors formulate AMR as a swarm reinforcement learning problem, in which multiple agents collaborate and share observations each other. By allowing the agents to be split into new agents in AMR process, a dense reward signal is efficiently feed to policy function to decide cells to refine. The experiments demonstrate that the method achieves competitive error against traditional error-based refinement strategies and generalizes well across different problem domains.

**Strengths:**

The paper is placed in the current growing literature on ML-based Adaptive Mesh Refinement (AMR), to enable scaling to large and complex systems. The authors propose formulating AMR as swarm reinforcement learning problem. The paper conducts various range of experiments and compare to RL-based baselines. The proposed method outperforms existing RL-based methods in most of the cases. The experiments are thorough, covering all the important ablations I could think of while reading the paper.

**Weaknesses:**

While the paper is generally well written, notations introduced in Section 3 are ambiguous. Below I list a few which made the paper harder to understand
* $M^{t,k}$ seems not to meet $\forall$ $j, $ $\sum M^{t, k}_{i, j} = 1$ since it seems to be defined by Hadmard product.
* Although definitions for state and observation are provided, they do not look well-defined; a local view of observation graph can have infinitely many choices.
* The definition of $J_{I}^{t}$ is also ambiguous; the domain of the policy comprises the observation and action spaces while reward is defined on state space and action space.
* What do $V$ and $E$ in the observation graph represent?

As someone not familiar with this field, I found the subsection on "Systems of Equations" in Section 4 described very clearly PDE systems defined on various types of (maybe complex) shapes. However it is still unclear to me how challenging solving AMR for those PDEs is, although they are claimed as being challenging in the introduction section. Can you expand on the details of PDEs and give some reasons that make solving AMR for those types of PDEs difficult?

For the evaluation of the wall-clock time of ASMR, I believe I don’t see plot of the proposed method in the figure in Appendix D.3. If so, it is difficult to compare the proposed method against other baselines.


**Questions:**

It would be interesting to see a study of how the hyperparameters of MPN (for policy function) impact the wall-clock time and error estimate. For example,
* Does large message passing steps decrease the error estimate and improves the scalability to much larger number of mesh elements?
* How much does the wall-clock time increase as increasing the message passing steps?


**Limitations:**

Yes, the paper discusses technical limitations in the main text.

---

> ### Author Rebuttal · Authors · 2023-08-09
>
> We sincerely appreciate the reviewer's constructive feedback. We are pleased to hear that they found our experiments and ablations to be conclusive and convincing. We also thank them for their remarks on the notations in Section 3 and their inquiries about the challenges of solving AMR for specific types of PDEs.
>
> > While the paper is generally well written, notations introduced in Section 3 are ambiguous. Below I list a few which made the paper harder to understand
> >
>
> We thank the reviewer for pointing out that the paper is currently unclear in terms of notation. We will clarify it during the revision, and briefly explain the notation in its current form in the following.
>
> > $M^{t,k}$ seems not to meet $\forall_j \sum M^{t,k}_{i,j}=1$ since it seems to be defined by Hadmard product.
> >
>
> $\mathbf{M}^{t,k}$ is defined by the (regular) matrix multiplication $\mathbf{M}^{t,k}=\mathbf{M}^t\mathbf{M}^{t+1}\dots\mathbf{M}^{t+k}$ in Line 150 in the paper. We will clarify that this is a matrix multiplication rather than a Hadamard product in the revision. As the individual matrices satisfy $\forall_j\sum_i \mathbf{M}_{ij}=1$ , so does their product.
>
> > Although definitions for state and observation are provided, they do not look well-defined; a local view of observation graph can have infinitely many choices.
> >
>
> The state of the system includes the mesh, its boundary conditions and its current solution. The observations encode this state as an observation graph (c.f. lines 166-169). The features for this graph depend on the current PDE, mesh and solution, which we detail in lines 222-229. We will adapt the paper to make the connection between the definition of the observation graph and its instantiation clearer. The local observation for each agent stems from its position in the observation graph. Our method is permutation equivariant due to the use of GNNs, ensuring consistent outputs from agents with identical local observations. We hope that this addresses the concern regarding the “infinitely many choices” and would appreciate the opportunity to provide further clarification if it does not.
>
> > The definition of $J_I^t$ is also ambiguous; the domain of the policy comprises the observation and action spaces while reward is defined on state space and action space.
> >
>
> The reward is defined as a function of state and action, while the policy is a function of the deterministic local observations induced by the mesh/graph. The missing link here is the observation function $\xi(s)$. The full equation above line 150 thus reads $J_i^t:=E_{\pi(\mathbf{a}|\xi(\mathbf{s}))}[\sum_{k=0}^{\infty}\gamma^k(\mathbf{M}^{t,k}\mathbf{r}(\mathbf{s}^{t+k},\mathbf{a}^{t+k}))_i]\text{,}$ which we will adapt in the revision.
>
> > What do $V$ and $E$ in the observation graph represent?
> >
>
> $V$ and $E$ are the vertices and edges in the observation graph respectively, and represent individual mesh elements and their neighborhood. This is briefly explained in Lines 166-167. We will clarify this notation in the revision.
>
> > Can you expand on the details of PDEs and give some reasons that make solving AMR for those types of PDEs difficult?
> >
>
> We thank the reviewer for the insightful question and the opportunity to expand on the details of the used PDEs. The general answer discusses which aspects make finding a good AMR strategy difficult for a particular PDE. On a very high level, finding a good refinement strategy for a PDE is difficult if there are certain areas of the domain that are much more interesting for the solution than others. A good AMR strategy must identify these areas and adapt their local resolution to a degree that depends on the local complexity of the problem
>
> > I believe I don’t see plot of the proposed method in the figure in Appendix D.3. If so, it is difficult to compare the proposed method against other baselines.
> >
>
> Regarding the evaluation of the wall-clock time of ASMR, Figure 10 on Page 22 of the appendix compares our method (represented by solid lines) with the calculation of the reference solution on a uniform refinement $\Omega^*$ (dashed lines). Other baselines are not part of this figure. The other RL-based methods with the exception of the *Argmax* baselines have a runtime that is comparable to ours due to their iterative refinement process but produce significantly worse refinements. Meanwhile, the local oracles use the reference mesh $\Omega^*$ which dominates their runtime. We will add a section in the revised version of the paper to clarify this relationship.
>
> > Does large message passing steps decrease the error estimate and improves the scalability to much larger number of mesh elements?
> >
>
> Our preliminary experiments with larger network architectures suggest a very small but consistent performance improvement when increasing the latent dimension from $32$ to $64$, whereas more than $2$ message passing steps do not significantly improve performance. For a Poisson problem with approximately $10\,000$ final mesh elements, the total runtime during inference increases by slightly over $10\,\%$ when either the latent dimension is increased to $64$ or the number of message passing steps is increased to $4$. While concrete numbers depend on the sizes of the individual meshes, executing the policy for the above example takes about $25$ to $30\,\%$ of the total runtime of a rollout, whereas calculating the solutions of the intermediate meshes takes up about $40$ to $45\,\%$ of time. The remaining time is taken up by the construction of the initial mesh and PDE and the calculation of the observation graph. We will provide more detailed results over different mesh resolutions in the revised version.
>
> We hope that our answer clarifies some of the details of our method. Should any issues persist or new ones arise, including ones regarding the notation in Section 3, we would greatly appreciate further communication.

---

### Official Review · Reviewer_fwaD · 2023-07-06

**Soundness:** 3 good
**Presentation:** 3 good
**Contribution:** 3 good
**Rating:** 6
**Confidence:** 3

**Summary:**

This paper presents a novel framework for adaptive mesh refinement for solving PDEs describing physical systems. The refinement is done as a Markov Decision Process in a Swarm RL setting, with each element of the mesh being an agent in the swarm. The agents' action space is binary and a learned policy decides whether an element should be refined or not. The system is supervised by a task-agnostic reward.

**Strengths:**

1. The problem of neural-AMR for efficient simulation is an important one, and the proposed solution is novel and promising.
2. The ability to have variable number of agents as the process progresses allows for finer refinements with more degrees of freedom.
3. The proposed refinement strategy doesn't require task-specific heuristics/rewards.
4. The improvement in speed as observed in the supplemental is impressive, especially for Fluid Flow.
5. Code is provided.

**Weaknesses:**

1. While the method handles complex PDEs, it does so only for 2D domains (as observed by the authors). it'd be interesting to see the results in 3D domains.

2. While Message Passing Networks do the job, it might be that Message Passing Attention Networks/ Graph Attention Networks as recently proposed might improve the results. I think an experiment in that direction is important to justify MPNs over more recent techniques.

**Questions:**

Can the method be extended/adapted for Mesh Movement instead of AMR? It'd extend the applicability of the proposed framework.

**Limitations:**

Same as those written in the weakness section - primarily, the results are only shown on 2D domains.

---

> ### Author Rebuttal · Authors · 2023-08-09
>
> We sincerely thank the reviewer for the valuable insight and suggestions, and particularly for highlighting the significance of our work's potential in addressing complex PDEs for efficient simulation. In the following, we want to address the individual points raised by the reviewer.
>
> > While the method handles complex PDEs, it does so only for 2D domains (as observed by the authors). it'd be interesting to see the results in 3D domains.
> >
>
> The present work focuses on 2D domains and conforming triangular meshes. We fully agree with the reviewer that extending the approach to 3D domains is an important next step, and briefly touch on the subject in the `Limitations and Future Work' section of the submission. While extending the experiments to 3D is out of the scope of this rebuttal period, we showcase the generalization capabilities of our method to significantly larger 2D domains in the PDF attached to the general answer.
>
> > Message Passing Attention Networks/ Graph Attention Networks as recently proposed might improve the results
> >
>
> We thank the reviewer for the suggestion. We use Message Passing Networks instead of e.g., Graph Attention Networks or Graph Convolutional Networks as they are the most general graph neural network architecture and have been used in related work for physical simulations (see e.g., Ref. 32, 37, 38) and learned adaptive mesh refinement (Ref. 27). The contributions of the present work lie in viewing mesh refinement as a swarm problem and providing spatial rewards and appropriate agent mappings for this problem. As such, the method is agnostic to the specific kind of GNN used. Still, we agree with the reviewer that a more optimized network architecture may improve downstream performance, which is important for practical applications of the presented method. We will add a discussion on the choice of network architecture to the revised paper.
>
> > Can the method be extended/adapted for Mesh Movement instead of AMR?
> >
>
> We thank the reviewer for the intriguing question. Mesh movement operations or R-Refinement could be implemented by extending the action space to include a velocity for each element’s barycenter and adapt the agent mapping based on overlapping element regions. Relatedly, we briefly discuss in lines 163-165 of the paper how the method can be extended to e.g., mesh coarsening operations. We find this to be a promising direction for future work, and provide a more detailed explanation for why this is not done in the present paper in the general answer.
>
> We hope that our responses address the reviewer's concerns, especially with respect to future extensions of the method. Should there be any further queries or unresolved issues, we encourage them to reach out to us.

---

### Official Review · Reviewer_Tf6s · 2023-07-07

**Soundness:** 2 fair
**Presentation:** 3 good
**Contribution:** 2 fair
**Rating:** 4
**Confidence:** 4

**Summary:**

The article presents an RL-based framework for adaptive mesh refinement in FEM. In this framework, each element in the mesh is perceived as an agent from RL's perspective. The observation for each agent is then constructed using a variety of local and global features relevant to the PDE of interest. The reward function is setup to account for the impact of agent's action in terms of error reduction and increase in compute cost, and also for future agents that are a result of a decision at current time (due to mesh refinement). The method is demonstrated to work for a variety of PDEs and be somewhat competitive with traditional error-based strategies.

**Strengths:**

Use of graph neural networks helps tackle unstructured meshes.
The agent mapping $M$ helps account for future agents that result from an agent due to mesh refinement.
Reward formulation helps account for error reduction and increase in compute cost due to mesh refinement.

**Weaknesses:**

While there is some novelty in terms of reward formulation and agent mapping, I find that the biggest weakness of the method is that it is only demonstrated to work for simple problems on meshes with only thousands of elements.

**Questions:**

None

**Limitations:**

The authors only talk about how the formulation can be used for de-refinement, but do not actually include it in their method.
The authors do not compare their method to existing estimators such as the one by Zienkiewicz-Zhu, which is fairly inexpensive and commonly used.
The demonstrated generalizations are for simple PDEs for relatively small meshes. Thus, I find the use of the phrase "complex simulations" in the abstract arguable.

---

> ### Author Rebuttal · Authors · 2023-08-09
>
> We are grateful to the reviewer for their insightful feedback, particularly on the significance of our reward formulation and agent mapping. We acknowledge their concerns about the scalability of our method and the necessity for a comparison with existing error estimators. In the following, we address each point raised by the reviewer in detail.
>
> > While there is some novelty in terms of reward formulation and agent mapping […]
> >
>
> We thank the reviewer for recognizing the innovative aspects of our work. We want to highlight that the novelty in our approach, particularly in reward formulation and agent mapping, plays a fundamental role in the success of our method. This is evidenced by the performance of our method when compared to the baselines, which mainly differ from our method in these key aspects.
>
> > The biggest weakness of the method is that it is only demonstrated to work for simple problems on meshes with only thousands of elements.
> >
>
> We provide new generalization experiments in our general answer and the accompanying PDF that show how ASMR generalizes to meshes with tens of thousands of elements when slightly adapting its training setup. The experiments in the main paper also show that existing RL-based methods struggle on the presented tasks, suggesting that the novel reward formulation and agent mapping in the present work mark an important step towards learning mesh refinement strategies for more complex problems.
>
> > The authors do not compare their method to existing estimators such as the one by Zienkiewicz-Zhu.
> >
>
> We sincerely thank the reviewer for suggesting to compare to additional established methods for AMR such as the Zienkiewicz-Zhu (ZZ) error estimator. We apply this error estimator to a percentage-based heuristic that refines all elements that are within the top $k\,\%$ of errors, and compare the results with ASMR on the Poisson problem in Figure 3 of the uploaded PDF.
>
> We find that the ZZ is significantly outperfomed by ASMR for the Poisson problem when applied as is. We find on preliminary visualizations that this is likely due to the ZZ error estimate missing entire modes of the load function. To compensate, we also compare to variants that uniformly refine the initial mesh one or two times (ZZ Error ({1,2}x Uniform)) before starting to apply the ZZ-error-based heuristic. When applying two initial refinements, the error estimate is competitive to ASMR on the top $0.1\,\%$ error, but significantly worse on the mean error metric. These results suggest that the ZZ error can provide competitive refinement when applied to suitable meshes, but that ASMR is preferable for general tasks. We will revise the paper to include the ZZ error estimate for all considered tasks.
>
> > The authors only talk about how the formulation can be used for de-refinement, but do not actually include it in their method.
> >
>
> We acknowledge that mesh de-refinement is important for general AMR strategies and aim to explore this direction further in future work. In the present submission, we focus on AMR for conforming triangular meshes and consider problems where de-refinement and coarsening operations are not crucial. We provide additional discussion in the general answer.
>
> > The method is demonstrated to work for a variety of PDEs and be somewhat competitive with traditional error-based strategies.
> >
>
> While traditional error-based strategies outperform our method in select scenarios, a considerable advantage of our approach lies in its enhanced computational speed. We show in Appendix D.3. that computing a reference mesh, which is required for the local oracles, is $2$ to $30$ times slower than our method. This advantage in computational speed is further amplified when considering larger meshes such as the one shown in Figure 2 of the uploaded PDF. Here, the refined mesh consists of slightly more than $50\,000$ elements, and computing it is more than $100$ times faster than generating a uniform refinement.
>
> We earnestly hope that our clarifications and provided materials address the concerns raised. If there are any lingering reservations or questions about the paper, we encourage the reviewer to reach out to us for further clarification.

---

### Author Rebuttal · Authors · 2023-08-09

We thank all reviewers for their valuable feedback on our submission. We are delighted to hear that the reviewers found our reward function and agent mapping innovative and useful (Tf6s), the speedup impressive (fwaD), the experiments convincing (ymhS), the improved stability of our method significant (LCc4), and the overall presentation to be of high quality and well-written (qjoP). We want to use this global response to address common concerns.

> Can the method generalize to larger and different problems? (Tf6s, LCc4, qjoP)

The most common concern raised by the reviewers was that of generalization to larger problems. We agree that it is crucial for the method to scale to larger (Tf6s, qjoP) and different (LCc4) problems and thus provide more significant speedups compared to the oracle heuristics.

In Appendix D.4, we illustrate our approach's effective generalization to new load functions and similarly sized domains. To generalize to larger domains, we adjust the observation space by removing global features (as suggested by qjoP), messages, and boundary node distances. We employ data augmentation: modifying training PDEs to mimic larger mesh segments by altering boundary conditions and load functions. The means of the load function are sampled from a centered unit Gaussian, allowing modes outside the mesh. We use domains with random holes (Appendix B.1) for varied initial meshes and apply random Gaussian loads to selected boundary parts as 'inlets'. See examples in Figure 1. We add an L2 norm of $3e$-$4$ to combat overfitting and omit the per-domain normalization of Lines 181-183. Due to time constraints, we only train for $200$ instead of $800$ iterations, reducing training time to a few hours on CPU. We emphasize that these changes only affect the observation space and the environment and leave the underlying algorithm unchanged.

We evaluate the resulting policy on a larger, spiral-shaped domain. This domain lies in $(0,20)^2$ compared to the $(0,1)^2$ domains seen during training and uses initial meshes with comparable element sizes. ASMR creates highly refined meshes with tens of thousands of elements, as showcased in Figure 2 in the attached PDF. The shown mesh has $50236$ elements, which is multiple times larger than any refinement shown by an existing RL-based method. Creating and solving this mesh takes $10$-$12$ seconds, whereas computing a uniform reference on the same domain takes over $20$ minutes and significantly more memory, or more than $100$ times as long. The error compared to the uniform mesh lies at $3$-$4\,\%$, depending on the metric, making it useful for most engineering applications.

We want to thank the reviewers for pointing us towards these findings and will incorporate these insights into the paper revision. We believe that these additional results significantly strengthen the claims of our paper and the applicability of our method.

> Mesh coarsening, 3-D domains and more complex problems are important for this line of work. (Tf6s, fwaD, qjoP)

While mesh coarsening is an important aspect of AMR, is it not essential for stationary problems with a unique solution. In the present submission, we prioritize conforming refinements due to their increased precision and numerical stability over non-conforming refinements [1]. For these refinement types, coarsening is complex due to dependencies on neighboring elements, as briefly discussed in lines 159-162. We do not consider 3D meshes here, as including them would not contribute to our core methodology but plan including them in future work. We choose problems from diverse fields to evaluate the methods across different solutions, but keep the underlying PDEs simple to ensure manageable computational demand. The difficulty of finding good refinements is evidenced by the comparatively poor performance of existing RL-based methods as mentioned in the previous answer.

> How does the method improve over the baselines, and why are the proposed tasks difficult? (Tf6s, ymhS, qjoP)

Our experiments include various elliptic PDEs on conforming triangular meshes with up to $6$ refinement steps across different domains. In contrast, existing work utilizes simpler domains and non-conforming quadrilateral meshes, often with only $2$ refinement levels (Ref. 27, 28) or problems that can be refined from local information (Ref. 29). The challenges in our tasks arise from the need for attention on multiple parts of the mesh (e.g., the Poisson problem) and pinpointed refinements (e.g., Linear Elasticity problem). This complexity is evidenced by the baseline methods significantly underperforming ASMR. Further, ASMR can provide high-quality refinements for significantly larger instances than any existing RL-based method, as shown in Figure 2 of the provided PDF. We recognize the need to differentiate between PDE complexity and refinement intricacy (see also [1]) and will address this in the revised paper.

> Why use a SwarMDP instead of a Dec-POMDP? (LCc4, qjoP)

We adapt the SwarMDP framework (Ref. 31) to agent-wise rewards, changing action and observation spaces to accommodate the varying number of agents, and crucially to mappings between these varying agents over time. While related work (Ref. 28) fits the varying action and observation spaces into a DEC-POMDP through the use of dummy states, viewing the mesh as a swarm system is conceptually simpler, makes the permutation-equivariance of the agents explicit, and allows for a more natural integration of both the agent-dependent reward and the mapping between agents. For the revision, we will add a brief discussion on this framework and its advantages.

We want to thank the reviewers again for their time and feedback and appreciate further communication if any concerns remain or appear during the discussion period.

[1] Nagarajan, A., & Soghrati, S. (2018). Conforming to interface structured adaptive mesh refinement: 3D algorithm and implementation. *Computational Mechanics*

---

### Author Response · Authors · 2023-08-21
**Summary of rebuttal period**

We again thank all reviewers for their time and effort in reviewing our paper, and appreciate that the majority vote for acceptance. Unfortunately, some of our rebuttals where left unanswered, so we would like to summarize the main concerns and lines of discussion during the review.

**********************************************************************Generalization experiments (Tf6s, LCc4, qjoP)**********************************************************************

We show in the rebuttal PDF that our method generalizes to significantly larger meshes on more complex domains during inference when slightly adapting the training setup. Here, the method provides accurate refinements and is more than $100$x faster than a uniform solution.

**Mesh coarsening and task complexity (Tf6s, fwaD, ymhS, qjoP)**

Our submission focuses on unstructured conforming meshes and predominantly stationary problems. Here, while coarsening is difficult to realize due to the non-local nature of refinements, it's typically redundant given the problem is stationary. Instead, the challenge for the policy is to find multiple precise refinement actions and manage the resulting large number of mesh elements. Our experiments show that while our method works well on this setting, the baselines break down for large meshes. This result strongly indicates that the experiments are at a higher level of complexity in terms of finding good refinement strategies than those considered in existing work. In the revision, we will clarify our emphasis on stationary problems and increased refinement steps to distinguish our paper from related work that considers non-stationary problems but simpler domains and shallow refinements.

**Zienkiewicz-Zhu error estimator (Tf6s)**

Results for the Zienkiewicz-Zhu error estimator for the Poisson problem are provided in the rebuttal PDF. We find that it performs poorly when applied out of the box. With additional tuning of the initial mesh, the method performs competitive with ours in terms of maximum remaining error, but is considerably worse on the mean error across the mesh.

**Baseline correctness (qjoP)**

We followed the implementations of the learned baselines as outlined in their original papers. Additionally, we implemented PPO and DQN variants of these baselines, keeping the core methodology and adapting it to a policy gradient and Q-learning setting respectively. We provide ablations over both algorithms and use the better of the two per method for all experiments. We adapted our “Argmax” baseline post-submission to more closely match the original paper, finding that its performance aligns with that in our initial submission. We use the same Message Passing Network architecture for all methods for comparability, and will include additional experiments on Graph Attention Networks in the revision.

We believe that the new results provided in the rebuttal PDF in combination with the clarification of the scope and complexity of our experiments opposed to previous work addresses the main concerns of the reviewers.

---

### Decision · Program_Chairs · 2023-09-21

**Decision:**

Accept (poster)

**Comment:**

This paper presents a fairly novel approach to handling variable number of nodes in the meshing procedure using MARL and is an interesting contribution to the community. Furthermore, the results are extensively benchmarked and demonstrate an interesting scaling to larger examples. However, the reviewers were quite concerned that statements about the performance relative to the baselines did not adequately reflect that the baselines were intended for non-stationary PDEs and changes to the wording in the paper should be made to reflect this. We expect the author to make the following changes to the final paper
- The large scale generalization example and results on ZZ should be included
- It should be made clearer what the relationship between the methods in [27, 28] and the methods labeled Argmax and VGDN are.
- The related work section should make clear that the compared baselines are intended to work on non-stationary PDEs and the limitations of the current work wrt other classes of PDEs should be clear.